# Inverse Entropic Optimal Transport Solves Semi-supervised Learning via Data Likelihood Maximization

## Abstract

Learning conditional distributions $\pi^*(\cdot|x)$ is a central problem in machine learning, which is typically approached via supervised methods with paired data $(x, y) \sim \pi^*$. However, acquiring paired data samples is often challenging, especially in problems such as domain translation. This necessitates the development of *semi-supervised* models that utilize both limited paired data and additional unpaired i.i.d. samples $x \sim \pi_x^*$ and $y \sim \pi_y^*$ from the marginal distributions. The usage of such combined data is complex and often relies on heuristic approaches. To tackle this issue, we propose a new learning paradigm that integrates both paired and unpaired data **seamlessly** using data likelihood maximization techniques. We demonstrate that our approach also connects intriguingly with inverse entropic optimal transport (OT). This finding allows us to apply recent advances in computational OT to establish a **light** learning algorithm to get $\pi^*(\cdot|x)$. In addition, we derive the universal approximation property demonstrating that our approach can theoretically recover true conditional distributions with arbitrarily small error. Furthermore, we demonstrate through empirical tests that our method effectively learns conditional distributions using paired and unpaired data simultaneously.

## 1 Introduction

Recovering conditional distributions $\pi^*(y|x)$ from data is one of the fundamental problems in machine learning, which appears both in predictive and generative modeling. In predictive modeling, the standard examples of such tasks are the classification, where $x \in \mathbb{R}^{D_x}$ is a feature vector and $y \in \{0, 1, \ldots, K\}$ is a class label, and regression, in which case $x$ is also a feature vector and $y \in \mathbb{R}^D$ is a real number. In generative modeling, both $x$ and $y$ are feature vectors in $\mathbb{R}^{D_x}, \mathbb{R}^{D_y}$, respectively, representing complex objects, and the goal is to find a transformation between them.

In our paper, we focus on the case when $x$ and $y$ are multi-dimensional real-value vectors and the true joint data distribution $\pi^*(x, y)$ is a continuous data distribution on $\mathbb{R}^{D_x} \times \mathbb{R}^{D_y}$, i.e., we exclude the problems when, e.g., $y$ is a discrete object such as the class label. That is, the scope of our paper is the multi-dimensional probabilistic regression problems, which can be referred to as **domain translation** problems, as usually $x$ and $y$ are feature vectors representing data from different domains. In turn, the goal is to make a (probabilistic) prediction, where for a new object $x_{new}$ from the input domain, we aim to predict the corresponding data $y(x_{new})$ from the target domain, according to the conditional distribution $\pi^*(y|x)$.

It is very natural that to learn the conditional distributions $\pi^*(y|x)$ of data one requires input-target data pairs $(x, y) \sim \pi^*$, where $\pi^*$ is the true joint distribution of data. In this case, $\pi^*(y|x)$ can be modeled via standard supervised learning approaches starting from a simple regression and ending with conditional generative models (Mirza & Osindero, 2014; Winkler et al., 2019). However, acquiring paired data may be costly, while getting unpaired samples $x \sim \pi_x^*$ or $y \sim \pi_y^*$ from two domains may be much easier and cheaper. This fact inspired the development of unsupervised (or unpaired) learning methods, e.g., (Zhu et al., 2017; Wu et al., 2020) among many others, which aim to somehow reconstruct the dependencies $\pi^*(y|x)$ with access to unpaired data only.

While both paired (supervised) and unpaired (unsupervised) domain translation approaches are being extremely well developed nowadays, surprisingly, the semi-supervised setup when **both** paired

and unpaired data is available is much less explored. This is due to the **challenge of designing learning objective** (loss) which can simultaneously take into account both paired and unpaired data. For example, one potential strategy here is to heuristically combine typical paired and unpaired losses. However, such a strategy leads to complex training objectives, see (Tripathy et al., 2019, §3.5), (Jin et al., 2019, §3.3), (Yang & Chen, 2020, §C), (Vasluianu et al., 2021, §3), (Panda et al., 2023, Eq. 8), (Tang et al., 2024, Eq. 8). Therefore, it is reasonable to raise a question: *is it possible to design a simple loss to learn $\pi^*(y|x)$ which **naturally** takes into account both paired and unpaired data?*

**Contributions.** In our paper, we positively answer the above-raised question. Namely,

1. We introduce a novel loss function (optimization objective) designed to facilitate the learning of conditional distributions $\pi^*(\cdot|x)$ using both paired and unpaired training samples derived from $\pi^*$ (§3.1). This loss function is based on the well-established principle of likelihood maximization. Our approach's notable advantage lies in its capacity to support end-to-end learning, thereby *seamlessly* integrating both paired and unpaired data into the training process.

2. We demonstrate the theoretical equivalence between our proposed loss function and the *inverse entropic optimal transport* problem (§3.2). This finding enables to leverage established computational OT methods to address challenges encountered in semi-supervised learning.

3. Building upon recent advancements in the field of computational optimal transport, we provide a *light* and *end-to-end* algorithm exploiting the Gaussian mixture parameterization specifically tailored to optimize our proposed likelihood-based loss function (in §3.3).

4. We prove that our proposed parameterization satisfies the universal approximation property. This finding theoretically allows our algorithm to recover $\pi^*$ arbitrarily well (§3.4).

Our empirical validation in §5 shows the impact of both unpaired and paired data on overall performance. In particular, our findings reveal that conditional distributions $\pi^*(\cdot|x)$ can be effectively learned even with a modest quantity of paired data $(x, y) \sim \pi^*$, provided that a sufficient amount of auxiliary unpaired data $x \sim \pi_x^*$, $y \sim \pi_y^*$ is available.

**Notations.** Throughout the paper, $\mathcal{X}$ and $\mathcal{Y}$ represent Euclidean spaces, equipped with the standard norm $\| \cdot \|$, induced by the inner product $\langle \cdot, \cdot \rangle$, i.e., $\mathcal{X} \stackrel{\text{def}}{=} \mathbb{R}^{D_x}$ and $\mathcal{Y} \stackrel{\text{def}}{=} \mathbb{R}^{D_y}$. The set of absolutely continuous probability distributions on $\mathcal{X}$ is denoted by $\mathcal{P}_{\text{ac}}(\mathcal{X})$. For simplicity, we use the same notation for both the distributions and their corresponding probability density functions. The joint probability distribution over $\mathcal{X} \times \mathcal{Y}$ is denoted by $\pi$ with corresponding marginals $\pi_x$ and $\pi_y$. The set of joint distributions with given marginals $\alpha$ and $\beta$ is represented by $\Pi(\alpha, \beta)$. We use $\pi(\cdot|x)$ for the conditional distribution, while $\pi(y|x)$ represents the conditional density at a specific point $y$. The differential entropy is given by $\mathrm{H}(\beta) = -\int_{\mathcal{Y}} \beta(y) \log \beta(y)\, dy$.

## 2 BACKGROUND

First, we recall the formulation of the domain translation problem (§2.1). We remind the difference between its paired, unpaired, and semi-supervised setups. Next, we recall the basic concepts of the inverse entropic optimal transport, which are relevant to our paper (§2.2).

### 2.1 DOMAIN TRANSLATION PROBLEMS

The goal of *domain translation* (DT) task is to transform data samples from the source domain to the target domain while maintaining the essential content or structure. This approach is widely used in applications like computer vision (Zhu et al., 2017; Lin et al., 2018; Peng et al., 2023), natural language processing (Jiang et al., 2021; Morishita et al., 2022), and audio processing (Du et al., 2022), etc. Domain translation task setups can be classified into supervised (or paired), unsupervised (or unpaired), and semi-supervised approaches based on the data used for training (see Figure 1).

**Supervised (Paired) Domain Translation** relies on matched examples from both the source and target domains, where each input corresponds to a specific output, enabling direct supervision during the learning process. Formally, this setup assumes access to a set of $P$ empirical pairs $XY_{\text{paired}} \stackrel{\text{def}}{=} \{(x_1, y_1), \dots, (x_P, y_P)\} \sim \pi^*$ from some unknown joint distribution. The goal here is to recover the conditional distributions $\pi^*(\cdot|x)$ to generate samples $y|x_{\text{new}}$ for new inputs $x_{\text{new}}$ that are not

present in the training data. While this task is relatively straightforward to solve, obtaining such paired training datasets can be challenging, as it often involves significant time, cost, and effort.

**Unsupervised (Unpaired) Domain Translation**, in contrast, does not require direct correspondences between the source and target domains (Zhu et al., 2017, Figure 2). Instead, it involves learning to translate between domains using unpaired data, which offers greater flexibility but demands advanced techniques to achieve accurate translation. Formally, we are given $Q$ unpaired empirical samples $X_{\text{unpaired}} \overset{\text{def}}{=} \{x_1, \ldots, x_Q\} \sim \pi_x^*$ from the source distribution and $R$ unpaired samples $Y_{\text{unpaired}} \overset{\text{def}}{=} \{y_1, \ldots, y_R\} \sim \pi_y^*$ from the target distribution. Our objective is to learn the conditional distributions $\pi^*(\cdot|x)$ of the unknown joint distribution $\pi^*$, whose marginals are $\pi_x^*, \pi_y^*$, respectively. Clearly, the primary challenge in unpaired setup is that the task is inherently ill-posed, leading to multiple potential solutions, many of which may be ambiguous or even not meaningful (Moriakov et al., 2020). Ensuring the translation's accuracy and relevance requires careful consideration of constraints and regularization strategies to guide the learning process (Yuan et al., 2018). Overall, the unpaired setup is very important because of large amounts of unpaired data in the wild.

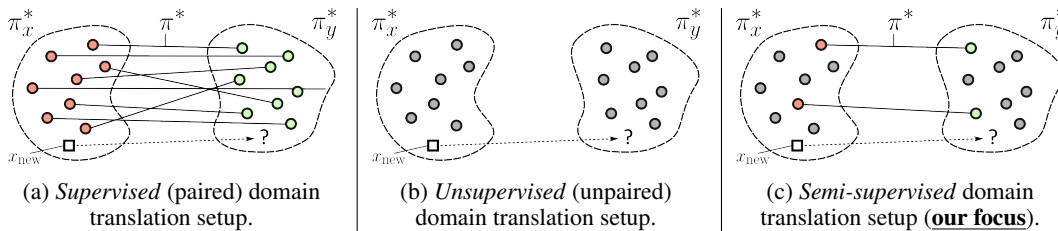

(a) *Supervised* (paired) domain translation setup.

(b) *Unsupervised* (unpaired) domain translation setup.

(c) *Semi-supervised* domain translation setup (**our focus**).

Figure 1: Visualization of domain translation setups. Red and green colors indicated paired training data $XY_{\text{paired}}$, while grey color indicates the unpaired training data $X_{\text{unpaired}}, Y_{\text{unpaired}}$.

**Semi-supervised domain translation** combines both approaches by utilizing a mix of paired and unpaired data (Tripathy et al., 2019; Jiang et al., 2023). This setup aims to leverage the advantages of paired data to guide the translation process while also taking advantage of the larger volume of unpaired data to improve the model's performance and generalization. Formally, one assumes access to both pairs $XY_{\text{paired}} \sim \pi^*$ and additional unpaired samples $X_{\text{unpaired}} \sim \pi_x^*$, $Y_{\text{unpaired}} \sim \pi_y^*$. Note that since paired samples can also be used in the unpaired manner, by convention, we assume that $P \leq Q, R$ and first $P$ unpaired samples are exactly the paired ones, i.e., $x_r' = x_r$ and $y_r' = y_r$ for $r \leq R$. In turn, the goal is still to learn $\pi^*(\cdot|x)$ using the available samples.

## 2.2 OPTIMAL TRANSPORT (OT)

The foundations of optimal transport (OT) are detailed in the seminal book by (Villani et al., 2009). For a more comprehensive overview, we refer to (Santambrogio, 2015; Peyré et al., 2019).

**Entropic OT** (Cuturi, 2013; Genevay, 2019). Consider source $\alpha \in \mathcal{P}_{\text{ac}}(\mathcal{X})$ and target $\beta \in \mathcal{P}_{\text{ac}}(\mathcal{Y})$ distributions. Let $c^* : \mathcal{X} \times \mathcal{Y} \to \mathbb{R}$ be a cost function. The *entropic* optimal transport problem between distributions $\alpha$ and $\beta$ is then defined as follows:

$$\text{OT}_{c^*, \varepsilon}(\alpha, \beta) \overset{\text{def}}{=} \min_{\pi \in \Pi(\alpha, \beta)} \mathbb{E}_{x, y \sim \pi}[c^*(x, y)] - \varepsilon \mathbb{E}_{x \sim \alpha} \text{H}(\pi(\cdot|x)), \tag{1}$$

where $\varepsilon > 0$ is the regularization parameter. Setting $\varepsilon = 0$ recovers the classic OT formulation (Villani et al., 2009) originally proposed by (Kantorovich, 1942). With mild assumptions, the transport plan $\pi^* \in \Pi(\alpha, \beta)$ that minimizes the objective (1) exists uniquely. It is called the *entropic OT plan*.

We note that in the literature, the entropy regularization term in (1) is usually $-\varepsilon \text{H}(\pi)$ or $+\varepsilon \text{KL}(\pi \| \alpha \times \beta)$. However, these forms are equivalent up to constants, see discussion in (Mokrov et al., 2024, §2) or (Gushchin et al., 2023, §1). In our paper, we work only with formulation (1), which is also known as the *weak* form of the entropic OT, see (Gozlan et al., 2017; Backhoff-Veraguas et al., 2019; Backhoff-Veraguas & Pammer, 2022).

**Dual formulation**. With mild assumptions on $c^*, \alpha, \beta$, the following dual OT formulation holds:

$$\text{OT}_{c^*, \varepsilon}(\alpha, \beta) = \sup_f \left\{ \mathbb{E}_{x \sim \alpha} f^{c^*}(x) + \mathbb{E}_{y \sim \beta} f(y), \right\} \tag{2}$$

where $f$ ranges over a certain subset of continuous functions (dual potentials) with mild assumptions on their boundness, see (Backhoff-Veraguas & Pammer, 2022, Eq. 3.3) for details. The term $f^{c^*}$ represents the so-called *weak entropic $c^*$-transform* of $f$, defined as:

$$f^{c^*}(x) \stackrel{\text{def}}{=} \min_{\beta \in \mathcal{P}(\mathcal{Y})} \left\{ \mathbb{E}_{y \sim \beta}[c^*(x, y)] - \varepsilon \mathrm{H}(\beta) - \mathbb{E}_{y \sim \beta} f(y) \right\}. \tag{3}$$

It has closed-form (Mokrov et al., 2024, Eq. 14), which is given by

$$f^{c^*}(x) = -\varepsilon \log \int_{\mathcal{Y}} \exp \left( \frac{f(y) - c^*(x, y)}{\varepsilon} \right) \mathrm{d}y. \tag{4}$$

**Inverse entropic OT.** The forward OT problem (1) focuses on determining the OT plan $\pi^*$ given a predefined cost function $c^*$. In contrast, the inverse problem provides the learner with a joint distribution $\pi^*$ and requires finding a cost function $c^*$ such that $\pi^*$ becomes the OT plan between its marginals, $\pi_x^*$ and $\pi_y^*$. This setup leads to the formulation of the *inverse entropic OT* problem, which can be expressed as the following minimization problem:

$$c^* \in \arg\min_c \left[ \underbrace{\left( \mathbb{E}_{x,y \sim \pi^*}[c(x, y)] - \varepsilon \mathbb{E}_{x \sim \pi_x^*} \mathrm{H}(\pi^*(\cdot|x)) \right)}_{\geq \mathrm{OT}_{c,\varepsilon}\left(\pi_x^*, \pi_y^*\right)} - \mathrm{OT}_{c,\varepsilon}\left(\pi_x^*, \pi_y^*\right) \right], \tag{5}$$

where $c$ skims through measurable functions $\mathcal{X} \times \mathcal{Y} \to \mathbb{R}$. The expression within the parentheses denotes the entropic transport cost of the plan $\pi^*$ in relation to the cost c between the marginals $\pi_x^*$ and $\pi_y^*$, thus ensuring that it is always greater than or equal to the optimal cost $\mathrm{OT}_{c,\varepsilon}(\pi_x^*, \pi_y^*)$. Consequently, the minimum achievable value for the entire objective is zero, which occurs only when $\pi^*$ corresponds to the optimal transport plan for the selected cost $c^*$. Here, the term $-\varepsilon \mathbb{E}_{x \sim \pi_x^*} \mathrm{H}(\pi(\cdot|x))$ can be omitted, as it does not depend on c. Additionally

- Unlike the forward OT problem (1), the entropic regularization parameter $\varepsilon > 0$ here plays no significant role. Indeed, by substituting $c(x, y) = \frac{\varepsilon}{\varepsilon'} c'(x, y)$ and multiplying the entire objective (5) by $\frac{\varepsilon'}{\varepsilon}$, one gets the inverse OT problem for $\varepsilon'$. Hence, the problems associated with different $\varepsilon$ are equivalent up to the change of variables, which is not the case for the forward OT (1).

- The inverse problem admits *several* possible solutions $c^*$. For example, $c^*(x, y) = -\varepsilon \log \pi^*(x, y)$ provides the minimum, which can be verified through direct substitution. Similarly, cost functions of the form $c^*(x, y) = -\varepsilon \log \pi^*(x, y) + u(x) + v(y)$ are also feasible, as adding terms dependent only on $x$ or $y$ does not alter the OT plan. In particular, when $u(x) = \varepsilon \log \pi_x^*(x)$ and $v(y) = 0$, one gets $c^*(x, y) = -\varepsilon \log \pi^*(y|x)$.

In practice, the joint distribution $\pi^*$ is typically available only through empirical samples, meaning that its density is often unknown. As a result, specific solutions such as $c^*(x, y) = -\varepsilon \log \pi^*(x, y)$ or $-\varepsilon \log \pi^*(y|x)$ cannot be directly utilized. Consequently, it becomes necessary to develop parametric estimators $\pi^\theta$ to approximate them using the available samples.

## 3 Semi-supervised Domain Translation via Inverse Entropic OT

In §3.1, we develop our proposed loss function that seamlessly integrates both paired and unpaired data samples. In §3.2, we demonstrate that derived loss is inherently linked to the inverse entropic optimal transport problem (5). In §3.3, we introduce lightweight parametrization to overcome challenges associated with optimizing the loss function. All our proofs can be found in Appendix C.

### 3.1 Loss Derivation

**Part I. Data likelihood maximization and its limitation.** Our goal is to approximate the true distribution $\pi^*$ by some parametric model $\pi^\theta$, where $\theta$ represents the parameters of the model. To achieve this, we would like to employ the standard KL-divergence minimization framework, also

known as data likelihood maximization. Namely, we aim to minimize

$$\text{KL}\left(\pi^*\|\pi^\theta\right) = \mathbb{E}_{x,y\sim\pi^*} \log \frac{\pi_x^*(x)\pi^*(y|x)}{\pi_x^\theta(x)\pi^\theta(y|x)} = \mathbb{E}_{x\sim\pi_x^*} \log \frac{\pi_x^*(x)}{\pi_x^\theta(x)} + \mathbb{E}_{x,y\sim\pi^*} \log \frac{\pi^*(y|x)}{\pi^\theta(y|x)} =$$

$$\text{KL}\left(\pi_x^*\|\pi_x^\theta\right) + \mathbb{E}_{x\sim\pi_x^*}\mathbb{E}_{y\sim\pi^*(\cdot|x)} \log \frac{\pi^*(y|x)}{\pi^\theta(y|x)} = \underbrace{\text{KL}\left(\pi_x^*\|\pi_x^\theta\right)}_{\text{Marginal}} + \underbrace{\mathbb{E}_{x\sim\pi_x^*}\text{KL}\left(\pi^*(\cdot|x)\|\pi^\theta(\cdot|x)\right)}_{\text{Conditional}}. \quad (6)$$

It is clear that objective (6) splits into two **independent** components: the *marginal* and the *conditional* matching terms. Our focus will be on the conditional component $\pi^\theta(\cdot|x)$, as it is the necessary part for the domain translation. Note that the marginal part $\pi_x^\theta$ is not actually needed. The conditional part of (6) can further be divided into the following two terms:

$$\mathbb{E}_{x\sim\pi_x^*}\mathbb{E}_{y\sim\pi^*(\cdot|x)}\left[\log\pi^*(y|x) - \log\pi^\theta(y|x)\right] = -\mathbb{E}_{x\sim\pi_x^*}\text{H}\left(\pi^*(\cdot|x)\right) - \mathbb{E}_{x,y\sim\pi^*}\log\pi^\theta(y|x). \quad (7)$$

The first term is independent on $\theta$, so we obtain the following minimization objective

$$\mathcal{L}(\theta) \stackrel{\text{def}}{=} -\mathbb{E}_{x,y\sim\pi^*}\log\pi^\theta(y|x). \quad (8)$$

It is important to note that minimizing (8) is equivalent to maximizing the conditional likelihood, a strategy utilized in conditional normalizing flows (Papamakarios et al., 2021, CNF). However, a major limitation of this approach is its reliance solely on paired data from $\pi^*$, which can be difficult to obtain in real-world scenarios. In the following section, we modify this strategy to incorporate available unpaired data within a semi-supervised learning setup (see §2.1).

**Part II. Solving the limitations via smart parameterization.** To address the above-mentioned issue and leverage unpaired data, we first use Gibbs-Boltzmann distribution density parametrization:

$$\pi^\theta(y|x) \stackrel{\text{def}}{=} \frac{\exp\left(-E^\theta(y|x)\right)}{Z^\theta(x)}, \quad (9)$$

where $E^\theta(\cdot|x) : \mathcal{Y} \to \mathbb{R}$ is *the Energy function*, and $Z^\theta(x) \stackrel{\text{def}}{=} \int_{\mathcal{Y}} \exp\left(-E^\theta(y|x)\right) \mathrm{d}y$ is the normalization constant (LeCun et al., 2006). Substituting (9) into (8), we get

$$-\mathbb{E}_{x,y\sim\pi^*}\log\pi^\theta(y|x) = \mathbb{E}_{x,y\sim\pi^*}E^\theta(y|x) + \mathbb{E}_{x\sim\pi_x^*}\log Z^\theta(x). \quad (10)$$

This objective already provides an opportunity to exploit the unpaired samples from the marginal distribution $\pi_x^*$ to learn the conditional distributions $\pi^\theta(\cdot|x) \approx \pi^*(\cdot|x)$. Namely, it helps to estimate the part of the objective related to the normalization constant $Z^\theta$. To incorporate separate samples from the second marginal distribution $\pi_y^*$, it is essential to choose a parametrization that allows to detach from the energy function $E^\theta(y|x)$ the term depending solely on $y$. Thus, we propose:

$$E^\theta(y|x) \stackrel{\text{def}}{=} \frac{c^\theta(x,y) - f^\theta(y)}{\varepsilon}. \quad (11)$$

The parameterization in (11) indeed permits the separation of the function $f^\theta(y)$. By setting $f^\theta(y) \equiv 0$ and $\varepsilon = 1$, the parameterization of the energy function $E^\theta(y|x)$ remains consistent, as it can be exclusively derived from $c^\theta(x,y)$. Finally, by substituting (11) into (10), we arrive at **our final objective**, which integrates both paired and unpaired data:

$$\mathcal{L}(\theta) = \underbrace{\varepsilon^{-1}\mathbb{E}_{x,y\sim\pi^*}[c^\theta(x,y)]}_{\text{Joint, requires pairs } (x,y)\sim\pi^*} - \underbrace{\varepsilon^{-1}\mathbb{E}_{y\sim\pi_y^*}f^\theta(y)}_{\text{Marginal, requires } x\sim\pi_y^*} + \underbrace{\mathbb{E}_{x\sim\pi_x^*}\log Z^\theta(x)}_{\text{Marginal, requires } x\sim\pi_x^*} \to \min_\theta. \quad (12)$$

At this point, a reader may come up with 2 reasonable questions regarding (12):

1. How to perform the optimization of the proposed objective? This question is not straightforward due to the existence of the (typically intractable) normalizing constant $Z_\theta$ in the objective.

2. To which extent do the separate terms in (12) (paired, unpaired data) contribute to the objective, and which type of data is the most important for learning the correct solution?

We answer these questions in §3.3 and §5. Before doing that, in the next section, we demonstrate a surprising finding that our proposed objective exactly solves the inverse entropic OT problem (5).

## 3.2 RELATION TO INVERSE ENTROPIC OPTIMAL TRANSPORT

In this section, we show that (5) is equivalent to (12). Indeed, directly substituting the dual form of entropic OT (2) into the inverse entropic OT problem (5) with the omitted entropy term yields:

$$\min_c \left\{ \mathbb{E}_{x,y \sim \pi^*}[c(x,y)] - \max_f \left[ \mathbb{E}_{x \sim \pi_x^*} f^c(x) + \mathbb{E}_{y \sim \pi_y^*} f(y) \right] \right\} = \tag{13}$$

$$\min_{c,f} \left\{ \mathbb{E}_{x,y \sim \pi^*}[c(x,y)] - \mathbb{E}_{x \sim \pi_x^*} f^c(x) - \mathbb{E}_{y \sim \pi_y^*} f(y) \right\}. \tag{14}$$

Now, let's assume that both $c$ and $f$ are parameterized as $c^\theta$ and $f^\theta$ with respect to a parameter $\theta$. Based on the definition provided in (4) and utilizing our energy function parameterization from (11), we can express $(f^\theta)^{c^\theta}(x)$ as follows:

$$(f^\theta)^{c^\theta}(x) = -\varepsilon \log \int_{\mathcal{Y}} \exp \left( \frac{f^\theta(y) - c^\theta(x,y)}{\varepsilon} \right) \mathrm{d}y = -\varepsilon \log Z^\theta(x). \tag{15}$$

This clarification shows that the expression in (13) aligns with our proposed likelihood-based loss in (12), scaled by $\varepsilon$. This finding indicates that *inverse entropic optimal transport (OT) can be interpreted as a likelihood maximization problem*, which opens up significant avenues to leverage established likelihood maximization techniques for optimizing inverse entropic OT, such as the evidence lower bound methods (Barber, 2012; Alemi et al., 2018) and expectation-maximization strategies (MacKay, 2003; Bishop & Bishop, 2023), etc.

Moreover, this insight allows us to reframe inverse entropic OT as *addressing the semi-supervised domain translation problem*, as it facilitates the use of both paired data from $\pi^*$ and unpaired data from $\pi_x^*$ and $\pi_y^*$. Notably, to our knowledge, the inverse OT problem has primarily been explored in *discrete* learning scenarios that assume access only to paired data (refer to §4).

## 3.3 PRACTICAL LIGHT PARAMETERIZATION AND OPTIMIZATION PROCEDURE

The most computationally intensive aspect of optimizing the loss function in (12) lies in calculating the integral for the normalization constant $Z^\theta$. To tackle this challenge, we propose a lightweight parameterization that yields closed-form expressions for each term in the loss function. Our proposed cost function parameterization $c^\theta$ is grounded in the LOG-SUM-EXP function (Murphy, 2012), which is widely recognized in the deep learning community for its practical advantages:

$$c^\theta(x,y) = -\varepsilon \log \sum_{m=1}^{M} v_m^\theta(x) \exp \left( \frac{\langle b_m^\theta(x), y \rangle}{\varepsilon} \right), \tag{16}$$

where $\{ v_m^\theta(x) : \mathbb{R}^{D_x} \to \mathbb{R}_+, b_m^\theta(x) : \mathbb{R}^{D_x} \to \mathbb{R}^{D_y} \}_{m=1}^{M}$ are arbitrary parametric functions, e.g., *neural networks*, with learnable parameters denoted by $\theta_c$. Inspired by the work (Korotin et al., 2024), we employ Gaussian mixture parametrization in the dual potential $f^\theta$:

$$f^\theta(y) = \varepsilon \log \sum_{n=1}^{N} w_n^\theta \mathcal{N}(y \mid a_n^\theta, \varepsilon A_n^\theta), \tag{17}$$

where $\theta_f \stackrel{\text{def}}{=} \{ w_n^\theta, a_n^\theta, A_n^\theta \}_{n=1}^{N}$ are learnable parameters of the potential, with $w_n^\theta \geq 0$, $a_n^\theta \in \mathbb{R}^{D_y}$, and $A_n^\theta \in \mathbb{R}^{D_y \times D_y}$ being a symmetric positive definite matrix. Thereby, our framework comprises a total of $\theta \stackrel{\text{def}}{=} \theta_f \cup \theta_c$ learnable parameters. For *clarity* and to *avoid notation overload*, we will omit the superscript $^\theta$ associated learnable parameters and functions in the subsequent formulas.

**Proposition 3.1** (Tractable form of the normalization constant). *Our parametrization of the cost function* (16) *and dual potential* (17) *delivers* $Z^\theta(x) \stackrel{\text{def}}{=} \sum_{m=1}^{M} \sum_{n=1}^{N} z_{mn}(x)$, *where*

$$z_{mn}(x) \stackrel{\text{def}}{=} w_n v_m(x) \exp \left( \frac{b_m^\top(x) A_n b_m(x) + 2 a_n^\top b_m(x)}{2\varepsilon} \right). \tag{18}$$

The proposition offers a closed-form expression for $Z^\theta(x)$, which is essential for optimizing (12). Furthermore, the following proposition provides a method for sampling $y$ given a new sample $x_{\text{new}}$.

**Proposition 3.2** (Tractable form of the conditional distributions). *From our parametrization of the cost function* (16) *and dual potential* (17) *it follows that the $\pi^\theta(\cdot|x)$ are Gaussian mixtures:*

$$\pi^\theta(y|x) = \frac{1}{Z^\theta(x)} \sum_{m=1}^{M} \sum_{n=1}^{N} z_{mn}(x) \mathcal{N}(y \,|\, s_{mn}(x), \varepsilon A_n), \tag{19}$$

*where $s_{mn}(x) \stackrel{\text{def}}{=} a_n + A_n b_m(x)$ and $z_{mn}(x)$ defined in Proposition 3.1.*

TRAINING. As stated in §2.1, since we only have access to the samples from the distributions, we will optimize the empirical counterpart of (12) via stochastic gradient descent in the parameters $\theta$:

$$\mathcal{L}(\theta) \approx \widehat{\mathcal{L}}(\theta) \stackrel{\text{def}}{=} \varepsilon^{-1} \frac{1}{P} \sum_{p=1}^{P} c^\theta(x_p, y_p) - \varepsilon^{-1} \frac{1}{R} \sum_{r=1}^{R} f^\theta(y_r) + \frac{1}{Q} \sum_{q=1}^{Q} \log Z^\theta(x_q) \to \min_\theta. \tag{20}$$

INFERENCE. According to our Proposition 3.2, the conditional distributions $\pi^\theta(\cdot|x)$ are Gaussian mixtures (19). As a result, sampling $y$ given $x$ is fast and straightforward.

### 3.4 UNIVERSAL APPROXIMATION OF THE LIGHT PARAMETERIZATION

One may naturally wonder how expressive is our proposed parameterization of $\pi_\theta$ in §3.3. Below we show that this parameterization allows approximating **any** distribution $\pi^*$ that satisfies mild compactness, boundness and regularity assumptions. We detail the assumptions in the proofs section.

**Theorem 3.1** (Light parameterization guarantees universal conditional distributions). *With mild assumptions on the joint distribution $\pi^*$, for all $\delta > 0$ there exists (a) an integer $N > 0$ and a Gaussian mixture $f^\theta$ (17) with $N$ components, (b) an integer $M > 0$ and (b.1) fully-connected neural networks $b_m^\theta : \mathbb{R}^{D_x} \to \mathbb{R}^{D_y}$ with ReLU activations and (b.2) fully-connected neural networks $v_m : \mathbb{R}^{D_x} \to \mathbb{R}_+$ with ReLU activations at hidden layers and with the exponential activation at the last layer such that $\pi^\theta$ defined by (9) and (11) satisfies $\mathrm{KL}\left(\pi^* \| \pi^\theta\right) < \delta$.*

We refer the reader to Theorem C.1 in Appendix C.2 for a precise formulation of Theorem 3.1.

## 4 RELATED WORKS

We review below the most related semi-supervised models and OT-based approaches to our work.

**Semi-supervised models.** As mentioned in §1, many existing semi-supervised domain translation methods combine paired and unpaired data by incorporating multiple loss terms into complex optimization objectives (Jin et al., 2019, §3.3), (Tripathy et al., 2019, §3.5), (Mustafa & Mantiuk, 2020, §3.2), (Paavilainen et al., 2021, §2), (Panda et al., 2023, Eq. 8), (Tang et al., 2024, Eq. 8). However, these approaches often require careful tuning of hyperparameters to balance the various loss terms.

We also note that there exist works addressing the question of incorpotating unpaired data to the log-likelihood training (8) by adding an extra likelihood terms, see CNFs-related works (Atanov et al., 2019; Izmailov et al., 2020). However, they rely on $x$ being a discrete object (e.g., a class label) and does not easily generalize to the continuous case, see Appendix B.2.2 for details.

The recent work by (Gu et al., 2023) utilizes both paired and unpaired data to build a transport plan based on key-point guided OT, initially introduced in (Gu et al., 2022). This transport plan is used as a heuristic to train a conditional score-based model on unpaired or semi-paired data. Overall, we note that the idea of applying OT in a semi-supervised manner traces back to the seminal work by (Courty et al., 2016), although their focus was on classification, not domain translation.

Another recent work by (Asadulaev et al., 2024) introduces a neural network-based OT framework for semi-supervised scenarios, utilizing general cost functionals for OT. However, their method requires **manually** constructing cost functions which can incorporate class labels or predefined pairs. In contrast, our approach adjusts the cost dynamically during training.

**(Inverse) OT solvers.** Our approach builds upon the light OT methods proposed by (Korotin et al., 2024; Gushchin et al., 2024), which introduce a *forward* solver for the entropic OT problem with the

quadratic cost function $c^*(x, y) = \frac{1}{2}\|x-y\|_2^2$ using the Gaussian Mixture parametrization. However, we consider a more general cost function (16) and incorporate cost function learning directly into the objective (20), in fact, producing an *inverse* OT (5) solver.

As highlighted in §2.2, the task of inverse optimal transport (IOT) implies learning the cost function from samples drawn from an optimal coupling $\pi^*$. Existing IOT solvers (Dupuy & Galichon, 2014; Dupuy et al., 2016; Li et al., 2019; Stuart & Wolfram, 2020; Ma et al., 2020; Chiu et al., 2022; Galichon & Salanié, 2022) focus on reconstructing cost functions from discrete marginal distributions, in particular, using the log-likelihood maximization techniques (Dupuy et al., 2016), see the introduction of (Andrade et al., 2023) for a review. In contrast, we develop a log-likelihood based approach aimed at learning a conditional distribution $\pi^\theta(\cdot|x) \approx \pi^*(\cdot|x)$ that incorporates both paired and unpaired data but not the cost function itself.

Recent work by (Howard et al., 2024) proposes a framework for learning cost functions to improve the mapping between the domains. However, it is limited by the use of deterministic mappings, i.e., does not have the ability to model non-degenerate conditional distributions.

## 5 EXPERIMENTAL ILLUSTRATIONS

We tested our solver on both synthetic data (§5.1) and real-world data distributions (§5.2). The code is written using the PyTorch framework and will be made publicly available. It is provided in the supplemental materials. Experimental details are given in Appendix B.

### 5.1 SWISS ROLL

**Setup.** For illustration purposes, we adapt the setup described in (Mokrov et al., 2024; Korotin et al., 2024) for our needs and consider a synthetic task where we transform samples from a Gaussian distribution $\pi_x^*$ into a Swiss Roll $\pi_y^*$ distribution (Figure 2a). The plan $\pi^*$ is generated by sampling from the mini-batch OT plan using the POT library (Flamary et al., 2021). We specifically chose a transportation cost for the minibatch OT to construct an optimal plan $\pi^*$ with bi-modal conditional distributions $\pi^*(\cdot|x)$ to assess how well our method performs in such a scenario. See Appendix B.2.1 for more details. During training, we use $P = 128$ paired (Figure 2b) and $Q = R = 1024$ unpaired samples. For an ablation study on how varying amounts of paired and unpaired data affect our method's performance, see Appendix B.2.4.

**Baselines.** We compare our method against several well-known generative modeling techniques, including: Conditional Normalizing Flow (Winkler et al., 2019, CNF), Conditional Generative Adversarial Network (Mirza & Osindero, 2014, CGAN), Unconditional GAN (Goodfellow et al., 2014) with $\ell^2$ loss supplement (UGAN+$\ell^2$), and just multilayer perceptron (MLP) regression with $\ell^2$ loss. Additionally, we show the results of the adaptation of semi-supervised log-likelihood-based losses by (Atanov et al., 2019; Izmailov et al., 2020), and denote the models by CNF (SS) and CGMM (SS), respectively, based on the paramterization used. For a detailed explanation of baselines employed in our experiments, please see Appendix B.2.2. Some of the baseline methods can fully utilize both paired and unpaired data during training, while others can use paired data only, see Table 1.

| Method | Paired $(x,y) \sim \pi^*$ | Unpaired $x \sim \pi_x^*$ | Unpaired $y \sim \pi_y^*$ |
|---|---|---|---|
| *Regression* | ✓ | ✗ | ✗ |
| *Conditional GAN* | ✓ | ✓ | ✗ |
| *Unconditional GAN + $\ell^2$* | ✓ | ✓ | ✓ |
| *Conditional NF* | ✓ | ✗ | ✗ |
| *Conditional NF (SS)* | ✓ | ✓ | ✓ |
| *Conditional GMM (SS)* | ✓ | ✓ | ✓ |
| ***Our method*** | ✓ | ✓ | ✓ |

Table 1: The ability to use of paired/unpaired data by various models.

**Discussion.** The results of the aforementioned methods are depicted in Figure 2. Clearly, the Regression model simply predicts the conditional mean $\mathbb{E}_{y \sim \pi^*(\cdot|x)} y$, failing to capture the full distribution. The CNF model suffers from overfitting, likely due to the limited availability of paired data $XY_{\text{paired}}$.

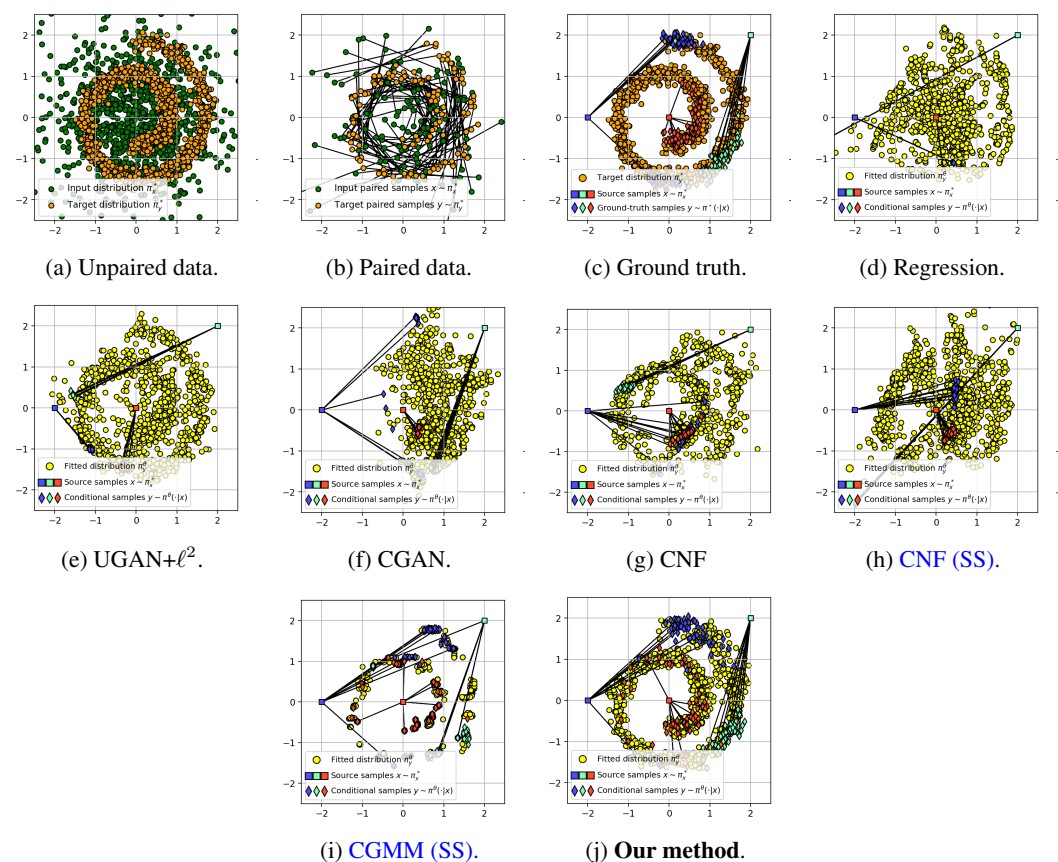

Figure 2: Comparison of the learned mapping on the *Gaussian → Swiss Roll* task. We use $P = 128$ paired data, $Q = 1024$ and $R = 1024$ unpaired source and target data, respectively.

The CGAN is unable to accurately learn the target distribution $\pi_y^*$, while the UGAN+$\ell^2$ fails to capture the underlying conditional distribution, resulting in suboptimal performance. The CNF (SS) does not provide improvement compared to CNF in this experiment, and CGMM (SS) model learns a degenerate solution, which is presumably due to the overfitting. As a sanity check, we evaluate all baselines using a large amount of paired data. Details are given in Appendix B.2.3.

## 5.2 WEATHER PREDICTION

Here we aim to evaluate our proposed approach on real-world data. We consider the *weather prediction* dataset (Malinin et al., 2021; Rubachev et al., 2024). The data is collected from weather stations and weather forecast physical models. It consists of $94$ meteorological features, e.g., pressure, wind, humidity, etc., which are measured over a period of one year at different spatial locations.

**Setup.** Initially, the problem was formulated as the prediction and uncertainty estimation of the air temperature at a specific time and location. We expand this task to the probabilistic prediction of all meteorological features, thereby reducing reliance on measurement equipment in remote and difficult-to-access locations, such as the Polar regions.

In more detail, we select two distinct months from the dataset and translate the meteorological features from the source month (January) to the target month (June). To operate at the monthly scale, we represent a source data point $x \in \mathbb{R}^{188}$ as the mean and standard deviation of the features collected at a specific location over the source month. The targets $y \in \mathbb{R}^{94}$ correspond to individual measurements in the target month. Pairs are constructed by aligning a source data point with the target measurements at the same location. Consequently, multiple target data points $y$ may correspond to a single source point $x$ and represent samples from conditional distributions $\pi^*(y|x)$. The

| | #unpaired | Baseline | | Ours | | | |
|---|---|---|---|---|---|---|---|
| #paired | | 0 | 10 | 50 | 100 | 250 | 500 |
| 10 | | $0.4 \pm .2$ | $17.9 \pm .3$ | $18.5 \pm .4$ | $18.4 \pm .2$ | $18.8 \pm .2$ | $19.2 \pm .3$ |
| 25 | | $3.5 \pm .09$ | $18.3 \pm .06$ | $18.7 \pm .2$ | $18.8 \pm .07$ | $19.5 \pm .1$ | $19.8 \pm .1$ |
| 50 | | $6.4 \pm .05$ | $18.7 \pm .2$ | $18.9 \pm .04$ | $19.2 \pm .2$ | $19.8 \pm .03$ | $20.3 \pm .4$ |
| 90 | | $6.5 \pm .1$ | $19 \pm .01$ | $19.4 \pm .05$ | $19.4 \pm .2$ | $20.3 \pm .05$ | $\mathbf{20.5} \pm .09$ |

Table 2: The values of the test *log-likelihood* $\uparrow$ on the *weather prediction* dataset obtained for a different number of paired and unpaired training samples.

| | Ours | CGAN | UGAN+$\ell^2$ | CNF | Regression | CGMM (SS) | CNF (SS) |
|---|---|---|---|---|---|---|---|
| log-likelihood$\uparrow$ | $\mathbf{20.5} \pm .09$ | N/A | N/A | $1.29 \pm .03$ | N/A | $0.32 \pm .03$ | $0.52 \pm .02$ |
| Conditional FD$\downarrow$ | $\mathbf{7.21} \pm .04$ | $15.79 \pm 1.11$ | $15.44 \pm 1.89$ | $18.72 \pm .09$ | $8.29 \pm .04$ | $7.17 \pm .07$ | $28.5 \pm .5$ |

Table 3: The values of the test *log-likelihood*$\uparrow$ and *Conditional Freshet distance*$\downarrow$ on the *weather prediction* dataset of our approach and baselines (trained on 500 unpaired and 90 paired samples).

measurements from non-aligned locations are treated as unpaired. We obtain 500 unpaired and 192 paired data samples. For testing, 100 pairs are randomly selected.

**Metrics and baselines**. We evaluate the performance of our approach by calculating the *log-likelihood* on the test target features. A natural baseline for this task is a probabilistic model that maximizes the likelihood of the target data. Thus, we implement an MLP that learns to predict the parameters of a mixture of Gaussians and is trained on the paired data only via the log-likelihood optimization (8). We also include all the baseline models from §5.1 trained on the available paired and unpaired data. Note that GAN models do not provide the density estimation and log-likelihood can not be computed for them. Therefore, we include the conditional Freshet distance metric. Namely, for each test $x$ we evaluate the Freshet distance (Heusel et al., 2017, Equation 6) between the predicted and the true features $y$. Then we average all these values obtained for all test inputs $x$.

**Results.** The results are presented in Tables 2 and 3. Result of Table 2 demonstrate that increasing the number of paired and unpaired data samples leads to improved test *log-likelihood*, which highlights the impact of the objective that employs both paired and unpaired data. Moreover, the proposed approach outperforms the baseline solution, which shows that even in problems where the paired data plays a key role for accurate predictions, incorporating the unpaired data can give an advantage. Additionally, the results in Table 3 confirm that our approach produces samples closer to the true distributions compared to the other baselines (with 500 unpaired and 90 paired samples).

# 6 DISCUSSION

**Limitations.** A limitation of our approach is that it uses the Gaussian Mixture parameterization for conditional distributions. This may limit its scalability. As a promising avenue for future work is incorporation of the more general parameterizations, such as neural networks, which are already well-studied in the context of forward entropic OT, see (Mokrov et al., 2024). In Appendix A, for completeness of the exposition, we showcase one possible way to use the neural parameterization for both cost and potential in our method via the energy-based modeling (LeCun et al., 2006, EBM).

**Potential impact.** Our framework has a simple and non-minimax optimization objective that seamlessly incorporates both unpaired and paired samples into the training. We expect that these advantages will encourage the use of our framework to develop other max-likelihood-based semi-supervised approaches based on more advanced (than Gaussian mixtures) techniques, e.g., energy-based models (LeCun et al., 2006; Du & Mordatch, 2019), diffusion models (Ho et al., 2020), etc.

**Broader impact.** This paper presents work whose goal is to advance the field of Machine Learning. There are many potential societal consequences of our work, none of which we feel must be specifically highlighted here.

**Reproducibility Statement** For all the presented experiments, a full set of hyperparameters is introduced either in §5 or in Appendix B. In addition, the code is submitted as supplementary material, with guidelines on how to run every experiment included.

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

## A    NEURAL PARAMETERIZATION

Throughout the main text, we parameterized the cost $c^\theta$ and potential $f^\theta$ using log-sum-exp functions and Gaussian mixtures (see §3.3). At this point, a reader may naturally wonder whether more general parameterizations for $c^\theta$ and $f^\theta$ can be used in our method, such as directly parameterizing both with neural networks. In this section, we affirmatively address this question by providing a procedure to optimize our main objective $\mathcal{L}(\theta)$ in (12) with general parameterizations for $c^\theta$ and $f^\theta$.

We note that a key advantage of our chosen parameterization (see §3.3) is that the normalizing constant $Z_\theta$ appearing in $\mathcal{L}(\theta)$ is available in the closed form. Unfortunately, this is not the case with general parameterizations of $c^\theta$ and $f^\theta$, necessitating the use of more advanced optimization techniques. While the objective $\mathcal{L}(\theta)$ itself may be intractable, we can derive its gradient, which is essential for optimization. The following proposition is derived in a manner similar to (Mokrov et al., 2024), who proposed methods for solving forward entropic OT problems with neural nets.

**Proposition A.1** (Gradient of our main loss (12))**.** *It holds that*

$$\frac{\partial}{\partial \theta}\mathcal{L}(\theta) = \varepsilon^{-1}\Bigg\{ \mathbb{E}_{x,y \sim \pi^*}\left[\frac{\partial}{\partial \theta}c^\theta(x,y)\right] - \mathbb{E}_{y \sim \pi_y^*}\left[\frac{\partial}{\partial \theta}f^\theta(y)\right]$$
$$+ \mathbb{E}_{x \sim \pi_x^*}\mathbb{E}_{y \sim \pi^\theta(y|x)}\left[\frac{\partial}{\partial \theta}\left(f^\theta(y) - c^\theta(x,y)\right)\right]\Bigg\}. \quad (21)$$

The formula for the gradient no longer includes the intractable normalizing constant $Z_\theta$. However, estimating the gradient requires sampling from the current model, i.e., obtaining $y \sim \pi^\theta(y|x)$. Unlike our Gaussian mixture-based parameterization (see §3.3), sampling from the model is more complex since we only have access to the unnormalized density of $\pi^\theta(y|x)$ through $c^\theta$ and $f^\theta$, and it is not necessarily a Gaussian mixture in this case. Nevertheless, this sampling can be accomplished using techniques for sampling from unnormalized densities, such as Markov Chain Monte Carlo (MCMC) methods (Andrieu et al., 2003). Thus, the gradient of the loss can be practically estimated, leading us to the following gradient-based training Algorithm 1.

---

**Algorithm 1:** Semi-supervised Learning via Energy-Based Modeling

**Input**  : Paired samples $XY_{\text{paired}} \sim \pi^*$; unpaired samples $X_{\text{unpaired}} \sim \pi_x^*$, $Y_{\text{unpaired}} \sim \pi_y^*$;
         potential network $f^\theta : \mathbb{R}^{D_y} \to \mathbb{R}$, cost network $c^\theta(x,y) : \mathbb{R}^{D_x} \times \mathbb{R}^{D_y} \to \mathbb{R}$;
         number of Langevin steps $K > 0$, Langevin discretization step size $\eta > 0$;
         basic noise std $\sigma_0 > 0$; batch size $N > 0$.
**Output:** trained potential network $f^{\theta^*}$ and cost network $c^{\theta^*}$ recovering $\pi^{\theta^*}(y|x)$ from (9).
**for** $i = 1, 2, \ldots$ **do**
   Derive batches $\{x_n\}_{n=1}^N = X \sim \pi_x^*$, $\{y_n\}_{n=1}^N = Y \sim \pi_y^*$, $\{\hat{x}_n, \hat{y}_n\}_{n=1}^N = XY \sim \pi^*$;
   Sample basic noise $Y^{(0)} \sim \mathcal{N}(0, \sigma_0)$ of size N;
   **for** $k = 1, 2, \ldots, K$ **do**
      Sample $Z^{(k)} = \{z_n^{(k)}\}_{n=1}^N$, where $z_n^{(k)} \sim \mathcal{N}(0,1)$;
      Obtain $Y^{(k)} = \{y_n^{(k)}\}_{n=1}^N$ with Langevin step:
      $y_n^{(k)} \leftarrow y_n^{(k-1)} + \frac{\eta}{2\varepsilon} \cdot \texttt{stop\_grad}\Big(\frac{\partial}{\partial y}\left[f^\theta(y) - c^\theta(x_n, y)\right]\big|_{y=y_n^{(k-1)}}\Big) + \sqrt{\eta}z_n^{(k)}$
   $\widehat{\mathcal{L}} \leftarrow \frac{1}{N}\Big[\sum_{x_n,y_n \in XY} c^\theta(x_n, y_n)\Big] - \frac{1}{N}\Big[\sum_{y_n \in Y} f_\theta(y_n)\Big] + \frac{1}{N}\Big[\sum_{y_n^{(K)} \in Y^{(K)}} f^\theta\left(y_n^{(K)}\right)\Big]$;
   Perform a gradient step over $\theta$ by using $\frac{\partial \widehat{\mathcal{L}}}{\partial \theta}$;

---

In the Algorithm 1, we employ the standard MCMC method called the Unadjusted Langevin Algorithm (ULA) (Roberts & Tweedie, 1996). For a detailed discussion on methods for training EBMs, refer to recent surveys (Song & Kingma, 2021; Carbone, 2024).

Overall, our proposed *inverse* OT algorithm turns to be closely related to the *forward* OT algorithm presented in (Mokrov et al., 2024, Algorithm 1). The key differences beside the obvious fact that

algorithms solve different problems are **(1)** we learn the cost function $c^\theta$ during the training process; **(2)** our learning exploits both paired and unpaired samples. Algorithms of this kind are usually called the **Energy-based models** (LeCun et al., 2006, EBM) because they parameterize the distributions of interest through their energy functions, i.e., minus logarithms of unnormalized densitites. Specifically, in the case of Algorithm 1, we learn unnormalized densities $\pi^\theta(y|x) \propto \exp\left(\frac{f^\theta(y) - c^\theta(x,y)}{\varepsilon}\right)$ defined through their energy functions $\varepsilon^{-1}(c^\theta(x,y) - f^\theta(y))$.

Below, we present an illustrative example on 2D data to demonstrate the ability of our Algorithm 1 to learn conditional plans using a fully neural network parametrization. We performed experiments on the Swiss roll matching problem (see §5.1) using two different datasets: one with 128 paired samples (as described in §5.1) and another with 16K paired samples (as detailed in Appendix B.2.3).

We employ MLPs with hidden layer configurations of $[128, 128]$ and $[256, 256, 256]$, using $LeakyReLU(0.2)$ for the parametrization of the potential $f^\theta$ and the cost $c^\theta$, respectively. The learning rates are set to $lr_{\text{paired}} = 5 \times 10^{-4}$ and $lr_{\text{unpaired}} = 2 \times 10^{-4}$. The sampling parameters follow those specified in (Mokrov et al., 2024).

It is worth noting that the model's ability to fit the target distribution is influenced by the amount of labeled data used during training. When working with partially labeled samples (as shown in Figure 3a), the model's fit to the target distribution is less accurate compared to using a larger dataset. However, even with limited labeled data, the model still maintains good accuracy in terms of the paired samples. On the other hand, when provided with fully labeled data (see Figure 3b), the model generates more consistent results and achieves a better approximation of the target distribution. A comparison of the results obtained using Algorithm 1 with neural network parametrization and those achieved using Gaussian parametrization (Figure 2j) reveals that Algorithm 1 exhibits greater instability. This observation aligns with the findings of (Mokrov et al., 2024, Section 2.2), which emphasize the instability and mode collapse issues commonly encountered when working with EBMs.

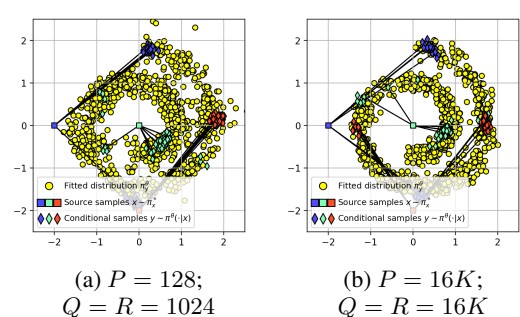

(a) $P = 128$; $Q = R = 1024$     (b) $P = 16K$; $Q = R = 16K$

Figure 3: Performance of our Algorithm 1 in the Swiss Roll mapping task (§5.1). We use MLPs to parametrize both the potential function $f^\theta$ and the cost function $c^\theta$.

**In conclusion,** it is important to recognize that the field of Energy-Based Models (EBMs) has undergone significant advancements in recent years, with the development of numerous scalable approaches. For examples of such progress, we refer readers to recent works by (Geng et al., 2024; Du et al., 2021; Gao et al., 2021) and other the references therein. Additionally, we recommend the comprehensive tutorial by (Song & Kingma, 2021) for an overview of training methods for EBMs. Given these advancements, it is reasonable to expect that by incorporating more sophisticated techniques into our basic Algorithm 1, it may be possible to scale the method to handle high-dimensional setups, such as image data. However, exploring these scaling techniques is beyond the scope of the current paper, which primarily focuses on the general methodology for semi-supervised domain translation. The investigation of methods to further scale our approach as a promising future research avenue.

# B    DETAILS OF THE EXPERIMENTS

## B.1    GENERAL IMPLEMENTATION DETAILS

To minimize (20), we parameterize $f^\theta$ (17) by representing $w_n$ using $\log w_n$, $a_n$ directly as a vector, and the matrix $A_n$ in diagonal form with $\log(A_n)_{i,i}$ on its diagonal. For $c^\theta$ (16), we parameterize $v_m(x)$ as a multilayer perceptron (MLP) (Haykin, 1998) with ReLU activations (Agarap, 2018) and a LogSoftMax output layer, while $b_m(x)$ is also modeled as an MLP with ReLU activations. The depth and number of hidden layers vary depending on the experiment.

To further simplify optimization, we use diagonal matrices $A_n$ in the parameterization of $f^\theta$, which not only significantly reduces the number of learnable parameters in $\theta_f$, but also enables efficient computation of $A_n^{-1}$ with a time complexity of $\mathcal{O}(D_y)$.

We utilize two separate Adam optimizers (Kingma, 2014) with different step sizes for paired and unpaired data to improve convergence.

As mentioned in §2.2, the solver is independent of $\varepsilon$, so we set $\varepsilon = 1$ for all experiments.

**Initialization.** We initialize $\log w_n$ as $\log \frac{1}{n}$, set $a_n$ using random samples from $\pi_y^*$, and initialize $\log(A_n)_{i,i}$ with $\log(0.1)$. For the neural networks, we apply the default PyTorch (Ansel et al., 2024) initialization.

## B.2 Gaussian To Swiss Roll Mapping

In all experiments conducted in this section, we set the parameters as follows: $N = 50$, $M = 25$, with learning rates $lr_\text{paired} = 3 \times 10^{-4}$ and $lr_\text{unpaired} = 0.001$. We utilize a two-layer MLP network for the function $b_m(x)$ and a single-layer MLP for $v_m(x)$. The experiments are executed in parallel on a 2080 Ti GPU for a total of 25,000 iterations, taking approximately 20 minutes to complete.

### B.2.1 Transportation cost matrix

To create the ground truth plan $\pi^*$, we utilize the following procedure: We start by sampling a mini-batch of size 64 and then determine the optimal mapping using the entropic Sinkhorn algorithm, as outlined in (Cuturi, 2013) and implemented in (Flamary et al., 2021). This process is repeated $P$ times to generate the required number of pairs.

We define the cost matrix for mini-batch OT as $C = \min(C^\varphi, C^{-\varphi})$, where $C^{\pm\varphi}$ represents matrices of pairwise $\ell_2$ distances between $x$ and $-y^{\pm\varphi}$, with $-y^{\pm\varphi}$ denoting the vector $-y$ rotated by an angle of $\varphi = \pm 90°$. In other words, $x \sim \pi_x^*$ maps to $y$ located on the opposite side of the Swiss Roll, rotated by either $\varphi$ or $-\varphi$, as shown in Figure 2c.

### B.2.2 Discussion of the Baselines

This section details the loss functions employed by the baseline models, providing context and explanation for the data usage summarized in Table 1. Furthermore, it explains a straightforward adaptation of the log-likelihood loss function presented in (8) to accommodate unpaired data, offering a natural comparative approach to the method proposed in our work.

- **Regression Model** (MLP) uses the following simple $\ell^2$ loss
$$\min_\theta \mathbb{E}_{(x,y)\sim\pi^*} ||y - G_\theta(x)||^2,$$

where $G_\theta : \mathcal{X} \to \mathcal{Y}$ is a generator MLP with trainable parameters $\theta$. Clearly, such a model can use only paired data. Furthermore, it is known that the optimal regressor $G^*$ coincides with $\mathbb{E}_{y\sim\pi^*(\cdot|x)}y$, i.e., predicts the conditional expectation. Therefore, such a model will never learn the true data distribution unless all $\pi^*(\cdot|x)$ are degenerate.

- **Conditional GAN** uses the following $\min\max$ loss:

$$\min_\theta \max_\phi \left[ \underbrace{\mathbb{E}_{x,y\sim\pi^*} \log\left(D_\phi(y|x)\right)}_{\text{Joint, requires pairs } (x,y) \sim \pi^*} + \underbrace{\mathbb{E}_{x\sim\pi_x^*}\mathbb{E}_{z\sim p_z(z)} \log\left(1 - D_\phi(G_\theta(z|x)|x)\right)}_{\text{Marginal, requires } x \sim \pi_x^*} \right],$$

where $G_\theta : \mathcal{Z} \times \mathcal{X} \to \mathcal{Y}$ is the conditional generator with parameters $\theta$, $p_z$ is a distribution on latent space $\mathcal{Z}$, and $D : \mathcal{Y} \times \mathcal{X} \to (0,1)$ is the conditional discriminator with parameters $\phi$. From the loss it is clear that the model can use not only paired data during the training, but also samples from $\pi_x^*$. The minimum of this loss is achieved when $G(\cdot|x)$ generates $\pi^*(\cdot|x)$ from $p_z$.

- **Unconditional GAN + $\ell^2$ loss** optimizes the following $\min\max$ objective:

$$\min_\theta \max_\phi \left[ \lambda \underbrace{\mathbb{E}_{(x,y)\sim\pi^*}\mathbb{E}_{z\sim p_z} ||y - G_\theta(x,z)||^2}_{\text{Joint, requires pairs } (x,y) \sim \pi^*} + \underbrace{\mathbb{E}_{y\sim\pi_y^*} \log\left(D_\phi(y)\right)}_{\text{Marginal, requires } y \sim \pi_y^*} + \underbrace{\mathbb{E}_{x\sim\pi_x^*}\mathbb{E}_{z\sim p_z} \log\left(1 - D_\phi(G_\theta(x,z))\right)}_{\text{Marginal, requires } x \sim \pi_x^*} \right],$$

where $\lambda > 0$ is a hyperparameter. In turn, $G_\theta : \mathcal{X} \times \mathcal{Z} \to \mathcal{Y}$ is the stochastic generator. Compared to the unconditional case, the main idea here is to use the unconditional disctiminator $D_\phi : \mathcal{Y} \to (0, 1)$. This allows using unpaired samples from $\pi_y^*$. However, using only GAN loss would not allow to use the paired information in any form, this is why the supervised $\ell^2$ loss is added ($\lambda = 1$). We note that this model has a trade-off between the target mathing loss (GAN loss) and regression loss (which suffers from averaging). Hence, the model is unlikely to learn the true paired data distribution and can be considered a heuristical loss to use both paired and unpaired data. Overall, we consider this baseline as most existing GAN-based solutions (Tripathy et al., 2019, §3.5), (Jin et al., 2019, §3.3), (Yang & Chen, 2020, §C), (Vasluianu et al., 2021, §3) for paired and unpaired data use objectives that are *ideologically* similar to this one.

- **Conditional Normalizing Flow** (Winkler et al., 2019) learns an explicit density model

$$\pi^\theta(y|x) = p_z(G_\theta^{-1}(y|x)) \left| \frac{\partial G_\theta^{-1}(y|x)}{\partial y} \right|$$

via optimizing log-likelihood (8) of the paired data. Here $G_\theta : \mathcal{Z} \times \mathcal{X} \to \mathcal{Y}$ is the conditional generator function. It is assumed that $\mathcal{Z} = \mathcal{Y}$ and $G_\theta(\cdot|x)$ is invertible and differentiable. In the implementation, we use the well-celebrated RealNVP neural architecture (Dinh et al., 2017). The optimal values are attached when the generator $G_\theta(\cdot|x)$ indeed generates $\pi^\theta(\cdot|x) = \pi^*(\cdot|x)$.

The conditional flow is expected to accurately capture the true conditional distributions, provided that the neural architecture is sufficiently expressive and there is an adequate amount of paired data available. However, as mentioned in §3.1, a significant challenge arises in integrating unpaired data into the learning process. For instance, approaches such as those proposed by (Atanov et al., 2019; Izmailov et al., 2020) aim to extend normalizing flows to a semi-supervised context. However, these methods primarily assume that the input conditions $x$ are discrete, making it difficult to directly apply their frameworks to our continuous case. For completeness, below we discuss a variant of the log-likelihood loss (Atanov et al., 2019, Eq. 1) when both $x, y$ are continuous.

- **Semi-supervised Conditional Normalizing Flows** (Atanov et al., 2019; Izmailov et al., 2020). As noted by the the authors, a natural strategy for log-likelihood semi-supervised training that leverages both paired and unpaired data is to optimize the following loss:

$$\max_\theta \left[ \underbrace{\mathbb{E}_{(x,y)\sim\pi^*} \log \pi^\theta(y|x)}_{\text{Joint, requires pairs } (x, y) \sim \pi^*} + \underbrace{\mathbb{E}_{y\sim\pi_y^*} \log \pi^\theta(y)}_{\text{Marginal, requires } y \sim \pi_y^*} \right]. \quad (22)$$

This straightforward approach involves adding the unpaired data component, $\mathbb{E}_{y\sim\pi_y^*} \log \pi^\theta(y)$ to the loss function alongside the standard paired data component (8). While loss (22) looks natural, its optimization is **highly non-trivial** since the marginal log-likelihood $\log \pi^\theta(y)$ is not directly available. In fact, (Atanov et al., 2019; Izmailov et al., 2020) use this loss *exclusively* in the case when $x$ is a discrete object, e.g., the class label $x \in \{1, 2, ..., K\}$ . In this case $\log \pi^\theta(y)$ can be analytically computed as the following finite sum

$$\log \pi^\theta(y) = \log \mathbb{E}_{x\sim\pi_x^*} \pi^\theta(y|x) = \log \sum_{k=1}^K \pi^\theta(y|x = k)\pi_x^*(x = k),$$

and $\pi^*(x = k)$ are known class probabilities. Unfortunately, in the continuous case $\pi_x^*(x)$ is typically not available explicitly, and one has to exploit **approximations** such as

$$\log \pi^\theta(y) = \log \mathbb{E}_{x\sim\pi_x^*} \pi^\theta(y|x) \approx \log \frac{1}{Q} \sum_{q=1}^Q \log \pi^\theta(y|x_q),$$

where $x_q$ are train (unpaired) samples. However, such Monte-Carlo estimates are generally **biased** (because of the logarithm) and do not lead to good results, especially in high dimensions. Nevertheless, for completeness, we also test how this approach performs. In our 2D example (Figure 2h), we found there is no significant difference between this loss and the fully supervised loss (8): both models incorrectly map to the target and fail to learn conditional distributions.

- **Semi-supervised Conditional Gaussian Mixture Model**. Using above-discussed natural loss (22) for semi-supervised learning, one may also consider a (conditional) Gaussian mixture parameterization for $\pi^\theta(y|x)$ instead of the conditional normalizing flow. For completeness of the

exposition, we also include such a baseline for comparison. For better transparency and fair comparison, we use the same Gaussian mixture paramteterization (19) as in our method. We found that such a loss quickly overfits to data and leads to degenerate solutions, see Figure 2i.

### B.2.3 BASELINES FOR SWISS ROLL WITH THE LARGE AMOUNT OF DATA (16K)

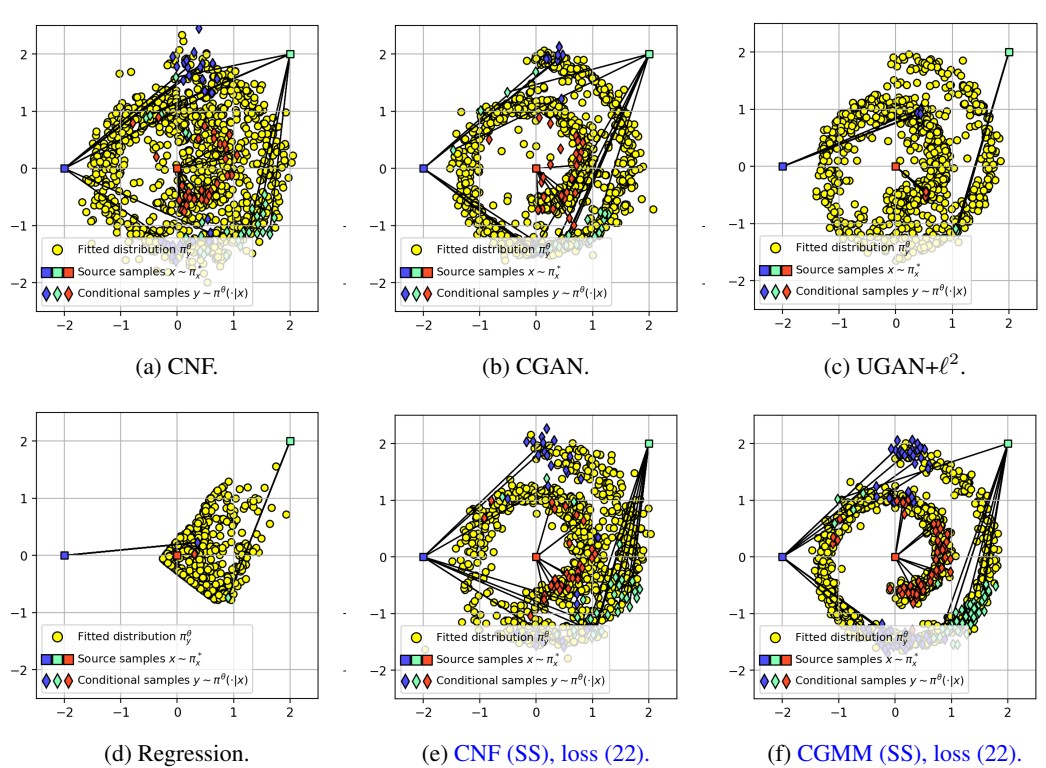

$$\begin{array}{ccc}\text{(a) CNF.} & \text{(b) CGAN.} & \text{(c) UGAN}+\ell^2.\end{array}$$

$$\begin{array}{ccc}\text{(d) Regression.} & \text{(e) CNF (SS), loss (22).} & \text{(f) CGMM (SS), loss (22).}\end{array}$$

Figure 4: Comparison of the mapping learned by baselines on *Gaussian → Swiss Roll* task (§5.1). We use $P = 16K$ paired data, $Q = R = 16K$ unpaired data for training.

In this section, we show the results of training of the baselines on the large amount of both paired (16K) and unpaired (16K) data (Figure 4). Recall that the ground truth $\pi^*$ is depicted in Figure 2c.

We see that given a sufficient amount of training data, Conditional GAN (Figure 4b) nearly succeeds in learning the true conditional distributions $\pi^*(\cdot|x)$. The same applies to the conditional normalizing flow (Figure 4a), but its results are slightly worse, presumably due to the limited expressiveness of invertible flow architecture. Regression, as expected fails to learn anything meaningful because of the averaging effect (Figure 4d). In turn, the unconditional GAN+$\ell^2$ (Figure 4c) nearly succeeds in generating the target data $\pi_y^*$, but the learned plan is incorrect because of the averaging effect.

Experiments using the natural semi-supervised loss function in (22) (Figure 4e) show that the loss function in (22) can reasonable well recover the conditional mapping with both CNF and CGMM parameterization, but it necessitates more training data than our proposed loss function (12). This conclusion is supported by the observation that CGMM model trained using (22) overfit, see (2i), whereas our method using objective (12) demonstrates good results, see Figure 2j.

### B.2.4 ABLATION STUDY

In this section, we conduct an ablation study to address the question posed in §3.1 regarding how the number of source and target samples influences the quality of the learned mapping. The results, shown in Figure 5, indicate that the quantity of target points $R$ has a greater impact than the number of source points $Q$ (compare Figure 5c with Figure 5b). Additionally, it is evident that the inclusion of unpaired data helps mitigate overfitting, as demonstrated in Figure 5a.

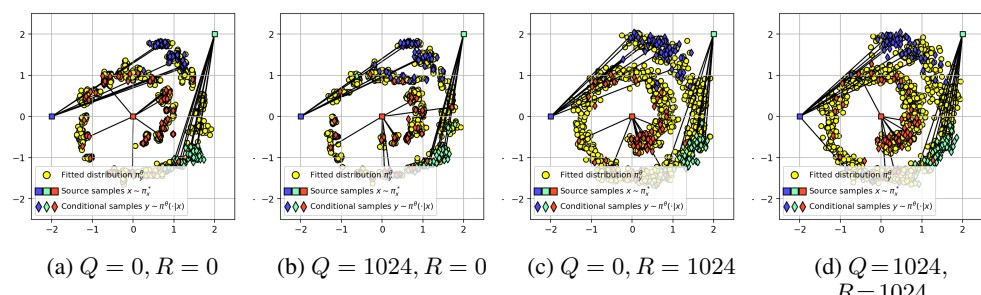

(a) $Q = 0, R = 0$    (b) $Q = 1024, R = 0$    (c) $Q = 0, R = 1024$    (d) $Q = 1024, R = 1024$

Figure 5: Ablation study analyzing the impact of varying source and target data point quantities on the learned mapping for the *Gaussian → Swiss Roll* task (using $P = 128$ paired samples).

### B.3 WEATHER PREDICTION

#### B.3.1 IMPLEMENTATION DETAILS

In general, we consider the same setting as in B. Specifically, we set $N = 10, M = 1$ and the number of optimization steps to $30,000$. The baseline uses an MLP network with the same number of parameters, predicting the parameters of a mixture of 10 Gaussians.

## C PROOFS

### C.1 FORMULAS FOR THE GAUSSIAN PARAMETRIZATION

*Proof of Proposition 3.1.* Thanks to our parametrization of the cost $c^\theta$ (16) and the dual potential $f^\theta$ (17), we obtain:

$$\exp\left(\frac{f^\theta(y) - c^\theta(x,y)}{\varepsilon}\right) = \exp\left(\log\sum_{n=1}^{N} w_n \mathcal{N}(y\,|\,a_n, \varepsilon A_n) + \log\sum_{m=1}^{M} v_m(x)\exp\left(\frac{\langle b_m(x), y\rangle}{\varepsilon}\right)\right)$$

$$= \sum_{m=1}^{M}\sum_{n=1}^{N} \frac{v_m(x)w_n}{\sqrt{\det\left(2\pi A_n^{-1}\right)}}\exp\left(-\frac{1}{2}(y - a_n)^\top\frac{A_n^{-1}}{\varepsilon}(y - a_n) + \frac{\langle b_m(x), y\rangle}{\varepsilon}\right)$$

Now we need to transform the expression above into the form of a Gaussian Mixture Model. To achieve this, we rewrite the formula inside the exponent using the fact that $A_n$ is a symmetric:

$$(y - a_n)^\top A_n^{-1}(y - a_n) - 2\langle b_m(x), y\rangle = y^\top A_n^{-1}y - 2a_n^\top A_n^{-1}y + a_n^\top A_n^{-1}a_n - 2\langle b_m(x), y\rangle =$$

$$y^\top A_n^{-1}y - 2\underbrace{\left(a_n + A_n b_m(x)\right)^\top}_{\stackrel{\text{def}}{=} s_{mn}^\top(x)} A_n^{-1}y + a_n^\top A_n^{-1}a_n =$$

$$(y - s_{mn}(x))^\top A_n^{-1}(y - s_{mn}(x)) + a_n^\top A_n^{-1}a_n - s_{mn}^\top(x)A_n^{-1}s_{mn}(x).$$

Afterwards, we rewrite the last two terms:

$$a_n^\top A_n^{-1}a_n - s_{mn}^\top(x)A_n^{-1}s_{mn}(x) = a_n^\top A_n^{-1}a_n - (a_n + A_n b_m(x))^\top A_n^{-1}(a_n + A_n b_m(x)) =$$

$$a_n^\top A_n^{-1}a_n - a_n^\top A_n^{-1}a_n - a_n^\top A_n^{-1}A_n b_m(x) - b_m^\top(x)A_n A_n^{-1}a_n - b_m^\top(x)A_n A_n^{-1}A_n b_m(x) =$$

$$-b_m^\top(x)A_n b_m(x) - 2a_n^\top b_m(x).$$

Finally, we get

$$\exp\left(\frac{f^\theta(y) - c^\theta(x,y)}{\varepsilon}\right) = \sum_{m=1}^{M}\sum_{n=1}^{N} \underbrace{w_n v_m(x) \exp\left(\frac{b_m^\top(x) A_n b_m(x) + 2a_n^\top b_m(x)}{2\varepsilon}\right)}_{\overset{\text{def}}{=} z_{mn}(x)}$$

$$\cdot \underbrace{\frac{1}{\sqrt{\det(2\pi A_n^{-1})}} \exp\left(-\frac{1}{2}(y - s_{mn}(x))^\top \frac{A_n^{-1}}{\varepsilon}(y - s_{mn}(x))\right)}_{= \mathcal{N}(y \,|\, s_{mn}(x), \varepsilon A_n)},$$

and thanks to $\int_{\mathcal{Y}} \mathcal{N}(y \,|\, s_{mn}(x), \varepsilon A_n)\mathrm{d}y = 1$, the normalization constant simplifies to the sum of $z_{mn}(x)$:

$$Z^\theta(x) = \int_{\mathcal{Y}} \exp\left(\frac{f^\theta(y) - c^\theta(x,y)}{\varepsilon}\right) \mathrm{d}y$$

$$= \int_{\mathcal{Y}} \sum_{m=1}^{M}\sum_{n=1}^{N} z_{mn}(x)\mathcal{N}(y \,|\, s_{mn}(x), \varepsilon A_n)\mathrm{d}y = \sum_{m=1}^{M}\sum_{n=1}^{N} z_{mn}(x).$$

$\square$

*Proof of Proposition 3.2.* Combining equations (9), (11) and derivation above, we seamlessly obtain the expression (19) needed for Proposition 3.2. $\square$

## C.2 UNIVERSAL APPROXIMATION

Our objective is to set up and use the very general universal approximation result in (Acciaio et al., 2024, Theorem 3.8). In what follows, we use the following notation. For any $d \in \mathbb{N}_+$ we denote the Lebesgue measure on $\mathbb{R}^d$ by $\lambda_d$, suppressing the subscript $d$ whenever clear from its context, we use $L_+^1(\mathbb{R}^d)$ to denote the set of Lebesgue integrable (equivalence class of) functions $f : \mathbb{R}^d \to \mathbb{R}$ for which $\int f(x)\,\lambda(dx) = 1$ and $f \geq 0$ $\lambda$-a.e; i.e. Lebesgue-densities of probability measures. We use $\mathcal{P}_1^+(\mathbb{R}^d)$ to denote the space of all Borel probability measures on $\mathbb{R}^d$ which are absolutely continuous with respect to $\lambda$, metrized by the total variation distance $d_{TV}$.

**Lemma 1** (The Space $(P_1^+(\mathbb{R}^d), d_{TV})$ is Quantizability by Gaussian Mixtures). *For every $q \in \mathbb{N}_+$, let $D_q \overset{\text{def.}}{=} \frac{q}{2}((d^2 + 3d + 2))$ and define the map*

$$Q_q : \mathbb{R}^{D_q} = \mathbb{R}^q \times \mathbb{R}^{qd} \times \mathbb{R}^{\frac{q}{2}d(d+1))} \to \mathcal{P}_1^+(\mathbb{R}^d)$$

$$\left(w, (\mu^{(i)})_{i=1}^q, (\Sigma^{(i)})_{i=1}^q\right) \mapsto \sum_{i=1}^{q} P_{\Delta_N}(w)_i \, \nu\left(\mu^{(i)}, \varphi(\Sigma^{(i)})\right)$$

*$P_{\Delta_N} : \mathbb{R}^q \mapsto \Delta_q$ is the $\ell^2$ orthogonal projection of $\mathbb{R}^q$ onto the $q$-simplex $\Delta_q \overset{\text{def.}}{=} \{u \in [0,1]^q : \sum_{i=1}^q u_i = 1\}$ and $\nu(\mu^{(i)}, \varphi(\Sigma^{(i)}))$ is the Gaussian measure on $\mathbb{R}^d$ with mean $\mu_i$, and non-singular covariance matrix given by $\varphi(\Sigma^{(i)})$ where for any $\Sigma \in \mathbb{R}^{d(d+1)/2}$ we define*

$$\varphi(\Sigma) \overset{\text{def.}}{=} \exp\left(\begin{pmatrix} \Sigma_1 & \Sigma_2 & \dots & \Sigma_d \\ \Sigma_2 & \Sigma_3 & \dots & \Sigma_{2d-1} \\ \vdots & \ddots & & \vdots \\ \Sigma_d & \Sigma_{2d-1} & \dots & \Sigma_{d(d+1)/2} \end{pmatrix}\right) \tag{23}$$

*where $\exp$ is the matrix exponential on the space of $d \times d$ matrices. Then, the family $(Q_q)_{q=1}^\infty$ is a quantization of $(P_1^+(\mathbb{R}^d), d_{TV})$ in the sense of (Acciaio et al., 2024, Definition 3.2).*

*Proof.* As implied by (Arabpour et al., 2024, Equation (3.10) in Proposition 7) every Gaussian measure $\mathcal{N}(m, \Sigma) := \mu$ on $\mathbb{R}^d$ with mean $m \in \mathbb{R}^d$ and symmetric positive-definite covariance matrix $\Sigma$ can be represented as

$$\mu = \mathcal{N}(m, \phi(X)) \tag{24}$$

for some (unique) vector $X \in \mathbb{R}^{d(d+1)/2}$. Therefore, by definition of a quantization, see (Acciaio et al., 2024, Definition 3.2), it suffices to show that the family of Gaussian mixtures is dense in $(\mathcal{P}_1^+(\mathbb{R}^d), d_{TV})$.

Now, let $\nu \in \mathcal{P}_1^+(\mathbb{R}^d)$ be arbitrary. By definition of $\mathcal{P}_1^+(\mathbb{R}^d)$ the measure $\nu$ admits a Radon-Nikodym derivative $f \stackrel{\text{def.}}{=} \frac{d\mu}{d\lambda}$, with respect to the $d$-dimensional Lebesgue measure $\lambda$. Moreover, by the Radon-Nikodym theorem, $f \in L_\mu^1(\mathbb{R}^d)$; and by since $\mu$ is a probability measure then $\nu \in L_+^1(\mathbb{R})$.

Since compactly-supported smooth functions are dense in $L_+^1(\mathbb{R}^d)$ then, for every $\varepsilon > 0$, there exists some $\tilde{f} \in C_c^\infty(\mathbb{R}^d)$ with $\tilde{f} \geq 0$ such that

$$\|f - \tilde{f}\|_{L^1(\mathbb{R}^d)} < \frac{\varepsilon}{3}. \tag{25}$$

Since $C_c^\infty(\mathbb{R}^d)$ is dense in $L^1(\mathbb{R}^d)$ then we may without loss of generality re-normalize $\tilde{f}$ to ensure that it integrates to 1. Since $\tilde{f}$ is compactly supported and approximates $f$, then (if $f$ is non-zero, which it cannot be as it integrates to 1) then it cannot be analytic, and thus it is non-polynomial. For every $\delta > 0$, let $\varphi_\delta$ denote the density of the $d$-dimensional Gaussian probability measure with mean 0 and isotropic covariance $\delta I_d$ (where $I_d$ is the $d \times d$ identity matrix). Therefore, the proof of (Pinkus, 1999, Proposition 3.7) (or any standard mollification argument) shows that we can pick $\delta \stackrel{\text{def.}}{=} \delta(\varepsilon) > 0$ small enough so that the convolution $\tilde{f} \star \varphi_\delta$ satisfies

$$\left\|\tilde{f} - \tilde{f} \star \varphi_\delta\right\|_{L^1(\mathbb{R}^d)} < \frac{\varepsilon}{3}. \tag{26}$$

Note that $\tilde{f} \star \varphi_\delta$ is the density of probability measure on $\mathbb{R}^d$; namely, the law of a random variable which is the sum of a Gaussian random variance with law $N(0, \delta I_N)$ and a random variable with law $\mu$. That is, $\tilde{f} \star \varphi_\delta \lambda \in L_+^1(\mathbb{R}^d)$. Together (25)and (26) imply that

$$\left\|f - \tilde{f} \star \varphi_\delta\right\|_{L^1(\mathbb{R}^d)} < \frac{2\varepsilon}{3}. \tag{27}$$

Recall the definition of the convolution: for each $x \in \mathbb{R}^d$ we have

$$\tilde{f}(x) \star \varphi_\delta \stackrel{\text{def.}}{=} \int_{u \in \mathbb{R}^d} \tilde{f}(u) \varphi_\delta(x - u) \lambda(du). \tag{28}$$

Since $\tilde{f}, \varphi_\delta \in C^\infty(\mathbb{R}^d)$ then Lebesgue integral of their product coincides with the Riemann integral of their product; whence, there is an $N \stackrel{\text{def.}}{=} N(\varepsilon) \in \mathbb{N}_+$ "large enough" so that

$$\left\|\int_{u \in \mathbb{R}^d} \tilde{f}(u) \varphi_\delta(x - u) \lambda(du) - \sum_{n=0}^N \tilde{f}(u_n) \varphi_\delta(x - u_n) \lambda(du)\right\|_{L^1(\mathbb{R}^d)} < \frac{\varepsilon}{3} \tag{29}$$

for some $u_1, \dots, u_N \in \mathbb{N}_+$. Note that, $\sum_{n=0}^N \tilde{f}(u_n) \varphi_\delta(x - u_n)$ is the law of a Gaussian mixture. Therefore, combining (27) and (30) implies that

$$\left\|f - \sum_{n=0}^N \tilde{f}(u_n) \varphi_\delta(x - u_n) \lambda(du)\right\|_{L^1(\mathbb{R}^d)} < \varepsilon. \tag{30}$$

Finally, recalling that the total variation distance between two measures with integrable Lebesgue density equals the $L^1(\mathbb{R}^d)$ norm of the difference of their densities; yields the conclusion; i.e.

$$d_{TV}(\nu, \hat{\nu}) = \left\|f - \sum_{n=0}^N \tilde{f}(u_n) \varphi_\delta(x - u_n) \lambda(du)\right\|_{L^1(\mathbb{R}^d)} < \varepsilon$$

where $\frac{d\hat{\nu}}{d\lambda} \stackrel{\text{def.}}{=} \sum_{n=0}^N \tilde{f}(u_n) \varphi_\delta(x - u_n) \lambda(du)$. □

**Lemma 2** (The space $(P_1^+(\mathbb{R}^d), d_{TV})$ is Approximate Simplicial). *Let $\hat{\mathcal{Y}} \stackrel{\text{def.}}{=} \cup_{N \in \mathbb{N}_+} \Delta_N \times [\mathcal{P}_1^+(\mathbb{R}^d)]^N$ and define the map $\eta : \hat{\mathcal{Y}} \mapsto \mathcal{P}_1^+(\mathbb{R}^d)$ by*

$$\eta(w, (\mu_n)_{n=1}^N) \stackrel{\text{def.}}{=} \sum_{n=1}^N w_n \mu_n.$$

*Then, $\eta$ is a mixing function, in the sense of (Acciaio et al., 2024, Definition 3.1). Consequentially, $(\mathcal{P}_1^+(\mathbb{R}^d), \eta)$ is approximately simplicial.*

*Proof.* Let $\mathcal{M}^+(\mathbb{R}^d)$ denote the Banach space of all finite signed measures on $\mathbb{R}^d$ with finite total variation norm $\|\cdot\|_{TV}$. Since $\|\cdot - \cdot\|_{TV} = d_{TV}$ when restricted to $\mathcal{P}_1^+(\mathbb{R}^d) \times \mathcal{P}_1^+(\mathbb{R}^d)$ and since $\|\cdot\|_{TV}$ is a norm, then the conclusion follows from (Acciaio et al., 2024, Example 5.1) and since $\mathcal{P}_1^+(\mathbb{R}^d)$ is a convex subset of $\mathcal{M}^+(\mathbb{R}^d)$. $\qquad\square$

Together, Lemmata 1 and 2 imply that $(\mathcal{P}_1^+(\mathbb{R}^d), d_{TV}, \eta, Q)$, where $Q \stackrel{\text{def.}}{=} (Q_q)_{q \in \mathbb{N}_+}$, is a QAS space in the sense of (Acciaio et al., 2024, Definition 3.4). Consequentially, the following is a geometric attention mechanism in the sense of (Acciaio et al., 2024, Definition 3.5)

$$\hat\eta : \cup_{N \in \mathbb{N}_+} \Delta_N \times \mathbb{R}^{N \times D_q} \to \mathcal{P}_1^+(\mathbb{R}^d)$$

$$\left( w, \left( w^{(n)}, (\mu^{(n:i)})_{i=1}^q, (\Sigma^{(n:i)})_{i=1}^q \right)_{n=1}^N \right) \mapsto \sum_{n=1}^N w_n \sum_{i=1}^q P_{\Delta_N}(w^{(n)})_i \, \nu\big(\mu^{(n:i)}, \varphi(\Sigma^{(n:i)})\big).$$

We are now ready to prove the first part of our approximation theorem.

**Proposition C.1** (Deep Gaussian Mixtures are Universal Conditional Distributions in the TV Distance)**.** *Let $\pi : (\mathbb{R}^d, \|\cdot\|_2) \to (\mathcal{P}_1^+(\mathbb{R}^d), d_{TV})$ be Hölder. Then, for every compact subset $K \subseteq \mathbb{R}^d$, every approximation error $\varepsilon > 0$ there exists $N, q \in \mathbb{N}_+$ and a ReLU MLP $\hat f : \mathbb{R}^d \mapsto \mathbb{R}^{N \times ND_q}$ such that the (non-degenerate) Gaussian-mixture valued map*

$$\hat\pi(\cdot|x) \stackrel{\text{def.}}{=} \hat\eta \circ f(x)$$

*satisfies the uniform estimate*

$$\max_{x \in K} d_{TV}\big(\hat\pi(\cdot|x) \| \pi(\cdot|x)\big) < \varepsilon.$$

*Proof.* Since Lemmata 2 and 1 imply that $(\mathcal{P}_1^+(\mathbb{R}^d), d_{TV}, \eta, Q)$, is a QAS space in the sense of (Acciaio et al., 2024, Definition 3.4), then the conclusion follows directly from (Acciaio et al., 2024, Theorem 3.8). $\qquad\square$

Since many of our results are formulated in the Kullback-Leibler divergence, then our desired guarantee is obtained only under some additional mild regularity requirements of the target conditional distribution $\hat\pi$ being approximated.

**Assumption 1** (Regularity of Conditional Distribution)**.** *Let $\pi : (\mathbb{R}^d, \|\cdot\|_2) \to (\mathcal{P}_1^+(\mathbb{R}^d), d_{TV})$ be Hölder, for each $x \in \mathbb{R}^d$, $\pi(\cdot|x)$ is absolutely continuous with respect to the Lebesgue measure $\lambda$ on $\mathbb{R}^d$, and suppose that there exist some $0 < \delta \le \Delta$ such that its conditional Lebesgue density satisfies*

$$\delta \le \frac{d\pi(\cdot|x)}{d\lambda} \le \Delta \qquad \text{for all } x \in \mathbb{R}^d. \tag{31}$$

**Theorem C.1** (Deep Gaussian Mixtures are Universal Conditional Distributions)**.** *Suppose that $\pi$ satisfies Assumption 1. Then, for every compact subset $K \subseteq \mathbb{R}^d$, every approximation error $\varepsilon > 0$ there exists $N, q \in \mathbb{N}_+$ such that: for each $n = 1, \dots, N$ and $i = 1, \dots, q$ there exist ReLU MLPs $w, v^{(n)}, \mu^{(n:i)}, \Sigma^{(n:i)}$ respectively mapping $\mathbb{R}^d$ to $\mathbb{R}^N$, $\mathbb{R}^d$, and $\mathbb{R}^{d(d+1)/2}$ such that the (non-degenerate) Gaussian-mixture valued map*

$$\hat\pi(\cdot|x) \stackrel{\text{def.}}{=} \sum_{n=1}^N P_{\Delta_N}(w(x))_n \sum_{i=1}^q P_{\Delta_N}(v^{(n)}(x))_i \, \nu\big(\mu^{(n:i)(x)}, \varphi(\Sigma^{(n:i)}(x))\big)$$

*satisfies the uniform estimate*

$$\max_{x \in K} d_{TV}\big(\pi(\cdot|x), \hat\pi(\cdot|x)\big) < \varepsilon. \tag{32}$$

*If, moreover, $\hat\pi$ also satisfies (31) (with $\hat\pi$ in place of $\pi$) then additionally*

$$\max_{x \in K} \mathrm{KL}\big(\pi(\cdot|x), \hat\pi(\cdot|x)\big) \in \mathcal{O}(\varepsilon), \tag{33}$$

*where $\mathcal{O}$ hides a constant independent of $\varepsilon$ and of the dimension $d$.*
**Moreover, the result holds even if each $\mu^{(n:i)}$ and $\Sigma^{(n:i)}$ are be assumed to be constant.**

The proof of Theorem C.1 makes use of the *symmetrized Kullback-Leibler divergence* $\mathrm{KL}_{sym}$ is defined for any two $\mu, \nu \in \mathcal{P}(\mathbb{R}^d)$ by $\mathrm{KL}_{sym}(\mu, \nu) \stackrel{\text{def.}}{=} \mathrm{KL}(\mu\|\nu) + \mathrm{KL}(\nu\|\mu)$; note, if $\mathrm{KL}_{sym}(\mu, \nu) = 0$ then $\mathrm{KL}_{sym}(\mu\|\nu) = 0$. We now prove our main approximation guarantee.

*Proof of Theorem C.1.* The first claim now directly follows from Proposition C.1 upon taking the MLPs $\mu^{(n:i)}$ and $\Sigma^{(n:i)}$ to be *constant* for all $i = 1, \ldots, q$ and $n = 1, \ldots, N$. It, therefore, only remains to establish the second estimate (33).

Under Assumption 1, $\pi(\cdot|x)$ and $\hat{\pi}(\cdot|x)$ are equivalent to the $d$-dimensional Lebesgue measure $\lambda$. Consequentially,

$$\pi(\cdot|x) \ll \hat{\pi}(\cdot|x)$$

for all $x \in \mathbb{R}^d$. Therefore, the Radon-Nikodym derivative $\frac{\hat{\pi}(\cdot|x)}{\pi(\cdot|x)}$ is a well-defined element of $L^1(\mathbb{R}^d)$, for each $x \in \mathbb{R}^d$; furthermore, we have

$$\frac{\pi(\cdot|x)}{\hat{\pi}(\cdot|x)} = \frac{\pi(\cdot|x)}{d\lambda} \frac{d\lambda}{\hat{\pi}(\cdot|x)}. \tag{34}$$

Again, leaning on Assumption 31 and the Hölder inequality, we deduce that

$$\sup_{a \in \mathbb{R}^d} \left| \frac{\pi(\cdot|x)}{\hat{\pi}(\cdot|x)}(a) \right| = \sup_{a \in \mathbb{R}^d} \left| \frac{\pi(\cdot|x)}{d\lambda}(a) \frac{d\lambda}{\hat{\pi}(\cdot|x)}(a) \right|$$

$$\leq \sup_{a \in \mathbb{R}^d} \left| \frac{\pi(\cdot|x)}{d\lambda}(a) \right| \sup_{a \in \mathbb{R}^d} \left| \frac{d\lambda}{\hat{\pi}(\cdot|x)}(a) \right|$$

$$\leq \sup_{a \in \mathbb{R}^d} \left| \frac{\pi(\cdot|x)}{d\lambda}(a) \right| \frac{1}{\delta}$$

$$\leq \frac{\Delta}{\delta} \tag{35}$$

where the final inequality held under the assumption that $\hat{\pi}$ also satisfies Assumption 31. Importantly, we emphasize that the right-hand side of (35) held *independently of $x \in \mathbb{R}^d$* ("which we are conditioning on"). A nearly identical estimate holds for the corresponding lower-bound. Therefore, we may apply (Sason, 2015, Theorem 1) to deduce that: there exists a constant $C > 0$ (independant of $x \in \mathbb{R}^d$ and depending only on the quantities $\frac{\Delta}{\delta}$ and $\frac{\delta}{\Delta}$; thus only on $\delta, \Delta$) such that: for each $x \in \mathbb{R}^d$

$$\mathrm{KL}\left(\pi(\cdot|x), \hat{\pi}(\cdot|x)\right) \leq C\, d_{TV}\left(\pi(\cdot|x), \hat{\pi}(\cdot|x)\right). \tag{36}$$

The conclusion now follows, since the right-hand side of (36) was controllable by the first statement; i.e. since (32) held we have

$$\mathrm{KL}\left(\pi(\cdot|x), \hat{\pi}(\cdot|x)\right) \leq C\, d_{TV}\left(\pi(\cdot|x), \hat{\pi}(\cdot|x)\right) \leq C\varepsilon. \tag{37}$$

A nearly identical derivation shows that

$$\mathrm{KL}\left(\hat{\pi}(\cdot|x), \pi(\cdot|x)\right) \leq C\varepsilon. \tag{38}$$

Combining (37) and (38) yields the following bound

$$\max_{x \in K} \mathrm{KL}_{sym}\left(\pi(\cdot|x), \hat{\pi}(\cdot|x)\right) \in \mathcal{O}(\varepsilon). \tag{39}$$

Since $\mathrm{KL}(\mu\|\nu) \leq \mathrm{KL}_{sym}(\mu, \nu)$ for every pair of Borel probability measures $\mu$ and $\nu$ on $\mathbb{R}^d$ then (39) implies (33). $\qquad\square$

## C.3 GRADIENT OF OUR LOSS FOR ENERGY-BASED MODELING

*Proof of Proposition A.1.* Direct differentiation of (12) gives:

$$\frac{\partial}{\partial \theta} \mathcal{L}(\theta) = \varepsilon^{-1} \mathbb{E}_{x,y \sim \pi^*} \left[ \frac{\partial}{\partial \theta} c^\theta(x, y) \right] - \varepsilon^{-1} \mathbb{E}_{y \sim \pi_y^*} \left[ \frac{\partial}{\partial \theta} f^\theta(y) \right] + \mathbb{E}_{x \sim \pi_x^*} \left[ \frac{\partial}{\partial \theta} \log Z^\theta(x) \right]. \tag{40}$$

Referring to equation (15) for the normalization constant, the last term can be expressed as follows:

$$\mathbb{E}_{x \sim \pi_x^*} \left[ \frac{1}{Z^\theta(x)} \frac{\partial}{\partial \theta} Z^\theta(x) \right] = \mathbb{E}_{x \sim \pi_x^*} \left[ \frac{1}{Z^\theta(x)} \int_{\mathcal{Y}} \frac{\partial}{\partial \theta} \exp \left( \frac{f^\theta(y) - c^\theta(x,y)}{\varepsilon} \right) \mathrm{d}y \right] =$$

$$\mathbb{E}_{x \sim \pi_x^*} \left[ \frac{1}{Z^\theta(x)} \int_{\mathcal{Y}} \frac{\frac{\partial}{\partial \theta} \left( f^\theta(y) - c^\theta(x,y) \right)}{\varepsilon} \exp \left( \frac{f^\theta(y) - c^\theta(x,y)}{\varepsilon} \right) \mathrm{d}y \right] =$$

$$\varepsilon^{-1} \mathbb{E}_{x \sim \pi_x^*} \left[ \int_{\mathcal{Y}} \frac{\partial}{\partial \theta} \left( f^\theta(y) - c^\theta(x,y) \right) \underbrace{\left\{ \frac{1}{Z^\theta(x)} \exp \left( \frac{f^\theta(y) - c^\theta(x,y)}{\varepsilon} \right) \right\}}_{\pi^\theta(y|x)} \mathrm{d}y \right].$$

From equation above we obtain:

$$\frac{\partial}{\partial \theta} \mathcal{L}(\theta) = \varepsilon^{-1} \Bigg\{ \mathbb{E}_{x,y \sim \pi^*} \left[ \frac{\partial}{\partial \theta} c^\theta(x,y) \right] - \mathbb{E}_{y \sim \pi_y^*} \left[ \frac{\partial}{\partial \theta} f^\theta(y) \right] \tag{41}$$

$$+ \mathbb{E}_{x \sim \pi_x^*} \mathbb{E}_{y \sim \pi^\theta(y|x)} \left[ \frac{\partial}{\partial \theta} \left( f^\theta(y) - c^\theta(x,y) \right) \right] \Bigg\}, \tag{42}$$

which concludes the proof. $\square$

