# OpenReview forum: "Inverse Entropic Optimal Transport Solves Semi-supervised Learning via Data Likelihood Maximization"
_ICLR.cc/2025/Conference — Submitted to ICLR 2025_

### Official Review · Reviewer_Nafh · 2024-11-01

**Soundness:** 3
**Presentation:** 2
**Contribution:** 3
**Rating:** 6
**Confidence:** 4

**Summary:**

The paper proposes a semi-supervised objective for the translation task that can leverage both paired and unpaired samples from the source and target domains.
The paper decomposes the standard MLE objective into three parts corresponding to a joint expectation and two marginal expectations over the unpaired source and target samples.
The key challenge is that parts of the objective require integration.
To avoid this computational challenge, the paper makes several critical assumptions about the joint cost function and the target-specific function that makes the necessary functions known in closed-form.
In particular, a Gaussian mixture model assumption and a simple log-sum-exp form of the cost function.
With these closed-form solutions, the paper then demonstrates the method on a 2D synthetic dataset and a weather prediction task.

**Strengths:**

- Proposes a promising MLE approach to incorporating both paired and unpaired data for the translation task.

- The proposed algorithm does not seem to have any complex hyperparameters.

- Shows a nice connection between using an energy-based conditional generative model and inverse entropic optimal transport. This opens up using insights and methods from computational OT.

- The paper was easy to read.

**Weaknesses:**

- Intuition and theoretic understanding of objective function is not clear. This makes it difficult to understand and extend. What is the purpose of the objectives corresponding to the Y and X marginals? Why does this make sense? What if these are misestimated?

- The assumptions are not clearly discussed and their corresponding limitations. What is the limitations of assuming the particular form of the needed parametric functions? It seems restrictive. Also, why is it required to have an energy-based model? For better or worse, score-based/diffusion-based generative models are far more common now. Is using an energy-based model significantly more restrictive?

- The practical motivation of the scenario is lacking. When would you expect this situation to hold in the real-world? Which real-world scenarios naturally have both paired and unpaired data?

- The experiments are relatively weak with only one 2D synthetic experiment and 1 weather experiment, but the weather experiment only includes a simple MLP regression baseline (i.e., none of the other baseline methods).

**Questions:**

- What is the intuitive interpretation of the new objective? This seems a bit ad-hoc/heuristic to introduce the $f(y)$ term for the sole purpose of using marginal samples from $p(y)$.

- Why use the parametrization in 16 and 17? What is the intuition of this parametrization? Is it a universal type of parametrization (i.e., can it represent all interesting cost functions)? Can you give examples of cost functions that do not have this form?

- How does this compare to the most natural likelihood objective that is $L(\theta) = E_{x,y}[-\log \pi^\theta(y|x)] + E_y[-\log E_x[\pi^\theta(y|x)]]$, which is basically the expected conditional KL plus a regularization term on the conditional that ensures it matches the marginal of p(y), i.e., integrating the conditional over the marginal of x should yield the marginal of y.  This seems to be a more straightforward (albeit hard-to-compute) semi-supervised objective.

- What does it mean that the x and y terms are asymmetric? It would seem that the marginal optimization terms should be symmetric as the unpaired data has no directionality per se.

- What if the paired data is not iid from the joint distribution? For example, if there are certain regions of the space that are more likely to have paired samples? This seems quite natural and reasonable in real-world scenarios. How would your method fair in these scenarios?

- How do you ensure $c^\theta(x,y)$ is a valid cost function?

*Summary of Review*
Overall, I found the paper to be intriguing, particularly in the connection between inverse optimal transport and the proposed objective. However, the derivations and development lacked strong intuition and explanation---rather seeming like mostly algebraic manipulation without a clear reason. The paper could be improved significantly with a much deeper understanding and interpretation of the objective. Furthermore, the assumptions lacked significant discussion while being relatively restrictive. Finally, the empirical results were relatively weak only having one 2D synthetic experiment and 1 weather experiment on a very small dataset with less than 1000 samples. Thus, the scalability and applicability of the method are questionable.

---

> ### Comment · Reviewer_Nafh · 2024-11-23
>
> Hi authors, I have read and reviewed your basic response. I think your response and edits did improve the paper somewhat though some of my concerns remain regarding intuition, deep understanding, and empirical validation. I will keep my score at this point.

---

> ### Author Response · Authors · 2024-11-23
> **Response to Reviewer Nafh (Part 1)**
>
> Dear reviewer, we thank you for already considering the general response to all the reviewers. Still we want to highlight that this answer does not take into account all your questions, so we provide a detailed answers to them below.
>
> **1. What is the intuitive interpretation of the new objective? This seems a bit ad-hoc/heuristic to introduce $f(y)$ the  term for the sole purpose of using marginal samples from $p(y)$.**
>
> Our objective is the direct log-likelihood maximization of data. In particular, introducing $f_{\theta}$ is theoretically justified because instead of just parameterizing $E^{\theta}(y|x)$ with a function of two variables (as it should be), we parameterize it as a function of two variables $c^{\theta}(x,y)$ plus a function of one variable $f^{\theta}(y)$. While this parameterization is more expressive (one may just set $f^{\theta}(y)=0$ and $c^{\theta}(x,y)=E^{\theta}(y|x)$), it allows to estimate the part of the loss using unpaired samples. Our ablation studies, e.g., **Table 2** (Real data) or **Appendix B.2.4** (Swiss Roll), show that this technique of variable separation indeed practically works and allows unpaired samples to provide notable assistance to paired ones.
>
> Also note that in the scenarios when the cost function is fixed (not learnable), our objective function (Equation 12) simplifies to the objective introduced [1, Equation 17], i.e., solving forward OT problem between unpaired samples for a given cost function.  This connection provides further justification for the inclusion of the $f^\theta(y)$ term and highlights the generality of our approach.
>
> **2. Intuition and theoretic understanding of objective <...> What is the purpose of the objectives corresponding to the Y and X marginals? Why does this make sense? What if these are misestimated?**
>
> Our loss is simply the KL divergence with a carefully designed energy-based reparameterization (Equation 9 and 11). The three components of the loss are **not separate** but rather integral parts of a single log-likelihood loss. Consequently, removing or misestimating any of these components would result in an incorrect estimation of the overall KL divergence.
>
> **3. Why use the parametrization in 16 and 17? What is the intuition of this parametrization? <...> Can you give examples of cost functions that do not have this form? <...> How do you ensure $c^\theta(x, y)$ is a valid cost function?**
>
> We selected this parametrization because it allows us to derive a closed-form solution for the normalization constant $Z_{\theta}$, which is essential for efficient computation. While it may be hard establish an intuition for such a parameterization, it is possible to explain how we come up with this parameterization. In fact, we analyzed the parametrization used in [2, 3] for the inner product OT cost $\langle x, y \rangle$, and found that the normalizing constant remains computable if one considers cost $\langle b_m(x), y \rangle$. Then we found that multiple such costs may be also smoothed with the *log-sum-exp* function without losing the necessary integration properties.
>
> Elaborating your question, we derived a new **Theorem 3.1** which shows that our proposed parameterization allows to universally approximate a broad class of conditional distributions (and, therefore, transport costs). Furthermore, in response to questions of the reviewers, we added an additional **Appendix A** showing that our framework supports arbitrary neural parametrizations.
>
> **4. Also, why is it required to have an energy-based model? <...> score-based/diffusion-based generative models are far more common now...**
>
> The main motivation for the energy-based model parametrization is because it allows us enables a clear separation between paired and the marginal distribution (Section 3.1). At the same time, we emphasize that according to work [4], Energy-based models can be viewed as a general framework encapsulating score and diffusion models. Therefore, it is highly expected that our ideas open the door to future extensions, such as incorporating score-based or diffusion models, but this is left for further research.
>
>  **5. How does this compare to the most natural likelihood objective that is $\mathcal{L}(\theta)=\mathbb{E}_{x, y \sim \pi^\star}[-\log\pi^\theta(y\vert x)] + \mathbb{E}_{y \sim \pi^\star_y}[-\log\mathbb{E}_{x \sim \pi^\star_x}[\pi^\star(y \vert x)]]$?**
>
> Thank you for highlighting this interesting loss. In fact, we already mentioned such approaches [5, 6] in Appendix B.2.2 of the original submission. As you correctly observe, this objective, while conceptually straightforward, presents computational challenges. Following you request, we added the detailed discussion of this approach to **Appendix B.2.2** and performed the comparison with it in the revised paper. As Figure 2i indicates, it overfits with limited data, whereas our method (Figure 2j) remains robust.

---

> > ### Author Response · Authors · 2024-11-23
> > **Response to Reviewer Nafh (Part 2)**
> >
> > **6. What does it mean that the $x$ and $y$ terms are asymmetric?**
> >
> > The asymmetry between $x$ and $y$ stems from our objective of learning the *conditional* distribution $\pi^\theta(y \vert x)$, which is inherently directional—predicting $y$ given $x$. Please note that we empirically studied this asymmetry through an ablation study, presented in **Appendix B.2.4** and visualized in **Figure 5**.
> >
> > **7. What if the paired data is not iid from the joint distribution? <...> How would your method fair in these scenarios?**
> >
> > Our approach is designed to handle the standard i.i.d. setup without imposing specific assumptions on the correlations of train samples. In turn, to estimate our loss in practice we use standard unbiased Monte-Carlo estimators using the available i.i.d. samples from $\pi^{\star}_x,\pi^{\star}_y, \pi^{\star}$. If the samples are non-i.i.d., it may be required to change the estimation scheme taking into account the correlations in data. This question of choosing the estimators for this case is interesting, but it is beyond the scope of our paper.
> >
> > **Concluding remarks.** Please respond to our post to let us know if the clarifications above suitably address your concerns about our work. We are happy to address any remaining points during the discussion phase; if the responses above are sufficient, we kindly ask that you consider raising your score.
> >
> > **References.**
> >
> > [1] Mokrov, Petr, Alexander Korotin, Alexander Kolesov, Nikita Gushchin, and Evgeny Burnaev. "Energy-guided Entropic Neural Optimal Transport." In The Twelfth International Conference on Learning Representations.
> >
> > [2] Korotin, Alexander, Nikita Gushchin, and Evgeny Burnaev. "Light Schr{\"o}dinger Bridge." In The Twelfth International Conference on Learning Representations.
> >
> > [3] Gushchin, Nikita, Sergei Kholkin, Evgeny Burnaev, and Alexander Korotin. "Light and Optimal Schr{\"o}dinger Bridge Matching." In Forty-first International Conference on Machine Learning. 2024.
> >
> > [4] Song, Yang, and Diederik P. Kingma. "How to train your energy-based models." arXiv preprint arXiv:2101.03288 (2021).
> >
> > [5] Atanov, Andrei, Alexandra Volokhova, Arsenii Ashukha, Ivan Sosnovik, and Dmitry Vetrov. "Semi-conditional normalizing flows for semi-supervised learning." arXiv preprint arXiv:1905.00505 920 (2019).
> >
> > [6] Izmailov, Pavel, Polina Kirichenko, Marc Finzi, and Andrew Gordon Wilson. "Semi-supervised learning with normalizing flows." In International conference on machine learning, pp. 4615-4630. PMLR, 2020.

---

> > > ### Author Response · Authors · 2024-11-24
> > > **Additional Clarification of Practical Applications**
> > >
> > > Dear reviewer, since you also had a question regarding the motivation of the semi-supervised domain translation setup, we decided to provide a separate answer listing some particular situations where this setup holds in the real world.
> > >
> > > **The practical motivation of the scenario is lacking. When would you expect this situation to hold in the real-world? Which real-world scenarios naturally have both paired and unpaired data?**
> > >
> > > The semi-supervised domain translation setup is highly relevant in many real-world domains, especially in cases where obtaining paired data is expensive or time-consuming, but unpaired data is readily available. This situation commonly arises in fields such as medical imaging, genomics, and electronic health records (EHR). We provide examples below.
> > >
> > > **1. Electronic Health Records (EHR):** EHR data often contains missing values, creating a scenario where some entries are complete (paired data) while others have missing fields (unpaired data). Semi-supervised methods can be applied to impute missing values by leveraging the structure of both paired and unpaired data, enhancing the completeness of EHR datasets. This, in turn, enables more comprehensive analyses and better clinical decision-making. The use of semi-supervised learning for imputing missing data in EHRs is an active research area (e.g., [1]).
> > >
> > > **2. Functional Brain Mapping (e.g., fMRI):** Pre-surgical planning using functional MRI (fMRI): fMRI scans can be separated into \emph{resting-state} and \emph{task-activated} types for the same individual. There are large datasets of unpaired resting-state and task-activated fMRI scans (Human Connectome Project (HCP) [2]). But collecting large datasets of the paired data can be problematic and such datasets are typically stored privately at clinics.
> > >
> > > **3. Single-Cell Genomics [3, 4]:** In single-cell RNA sequencing studies, understanding the transition between primary tumors and metastases often requires paired data from the same patient at both stages, which is rare due to the difficulty of obtaining such samples. However, there are large unpaired datasets of single-cell expression profiles from primary tumors and metastases across different patients.
> > >
> > > **Clarification on Scope:** While these examples illustrate the practical motivation and relevance of scenarios involving both paired and unpaired data, we do not specifically consider these applications in our work due to a lack of domain expertise and the proprietary nature of many relevant datasets. Our study focuses on the general methodology, which could potentially be adapted to these domains with further investigation and domain-specific modifications.
> > >
> > > **References.**
> > >
> > > [1] Beaulieu-Jones, Brett K., and Casey S. Greene. "Semi-supervised learning of the electronic health record for phenotype stratification." Journal of biomedical informatics 64 (2016): 168-178.
> > >
> > > [2] Van Essen, David C., Stephen M. Smith, Deanna M. Barch, Timothy EJ Behrens, Essa Yacoub, Kamil Ugurbil, and Wu-Minn HCP Consortium. "The WU-Minn human connectome project: an overview." Neuroimage 80 (2013): 62-79.
> > >
> > > [3]  Haque, Ashraful, Jessica Engel, Sarah A. Teichmann, and Tapio Lönnberg. "A practical guide to single-cell RNA-sequencing for biomedical research and clinical applications." Genome medicine 9 (2017): 1-12.
> > >
> > > [4] Jovic, Dragomirka, Xue Liang, Hua Zeng, Lin Lin, Fengping Xu, and Yonglun Luo. "Single‐cell RNA sequencing technologies and applications: A brief overview." Clinical and translational medicine 12, no. 3 (2022): e694.

---

> > > > ### Author Response · Authors · 2024-11-28
> > > > **Official Comment by Authors**
> > > >
> > > > Dear Reviewer Nafh,
> > > >
> > > > Thank you for the valuable feedback on our work. This is a gentle reminder that the discussion phase concludes in less than three working days, specifically at 11:59 PM AoE on December 2.
> > > >
> > > > We are eager to address any additional questions you may have. If you feel that our responses have adequately addressed your concerns, we kindly ask you to consider increasing your score.
> > > >
> > > > Your thoughtful comments and time are greatly appreciated.

---

> > > > > ### Comment · Reviewer_Nafh · 2024-12-02
> > > > >
> > > > > Hi authors,
> > > > >
> > > > > Thank you for your additional response. These helped answer some of my questions. I'm still concerned about the deep understanding and explanation of the objective (or at least with clarity w.r.t. the development). In addition, the empirical results are still relatively minimal, which seemed unaddressed in the response. Nonetheless, I still lean towards acceptance as I did at first so I will keep my score as borderline accept.

---

### Official Review · Reviewer_vGg8 · 2024-11-04

**Soundness:** 3
**Presentation:** 3
**Contribution:** 2
**Rating:** 5
**Confidence:** 4

**Summary:**

This paper presents a novel approach to semi-supervised domain translation by introducing a likelihood-based loss function that naturally incorporates both paired and unpaired samples. The key theoretical contribution is establishing a connection between their loss and inverse entropic optimal transport - a connection that hasn't been previously explored in the literature. The authors propose using a Gaussian mixture parameterization that allows closed-form expressions for their loss terms. They empirically validate their method against standard supervised/unsupervised approaches on synthetic 2D data and a weather prediction task.

While the paper presents an interesting theoretical contribution connecting inverse entropic optimal transport with likelihood estimation, the lack of comparisons against competing semi-supervised methods and limited experimental validation makes it difficult to assess the practical value of the proposed approach. Thus, currently the paper does not reach the bar for acceptance.

**Strengths:**

- Novel likelihood maximization loss for semi-supervised domain translation that elegantly handles both paired and unpaired data within a single objective.
- Novel theoretical connection established between inverse entropic optimal transport and likelihood estimation.
- Empirical validation demonstrating how performance scales with both paired and unpaired sample sizes.

**Weaknesses:**

- Experimental validation:
	- While the related work section discusses multiple approaches to semi-supervised domain translation, the paper lacks empirical comparisons against any of these existing methods. This makes it difficult to assess the relative advantages of the proposed approach compared to established methods that also leverage both paired and unpaired data.
	- The paper would be substantially strengthened by: 1) clearly articulating the advantages of the proposed method over existing approaches, and 2) providing direct empirical comparisons against state-of-the-art semi-supervised methods. Currently, comparisons are limited to baseline methods that were not specifically designed for semi-supervised domain translation.
	- The experimental validation would benefit from including tests on higher-dimensional problems, such as the image translation tasks explored in [1], to demonstrate the method's scalability beyond simple domains.
- Reliance on Gaussian mixture parameterization:
	- The authors acknowledge that their choice of Gaussian mixture parameterization inherently limits the method's scalability to more complex domains.
	- The discussion of alternative parameterizations seems incomplete, leaving questions about potential extensions to more expressive models.
	- While the authors claim their parameterization is "light-speed," this assertion lacks proper justification or comparative analysis.

[1] Xiaole Tang, Xin Hu, Xiang Gu, and Jian Sun. "Residual-conditioned optimal transport: Towards structure-preserving unpaired and paired image restoration". ICML 2024.

**Questions:**

- The authors describe their parameterization as "light-speed" - what are the specific computational advantages that justify this claim?
- What makes Gaussian mixture parameterization particularly suitable for the proposed loss compared to alternative parameterizations?
- How does the proposed method compare to existing semi-supervised domain translation techniques in terms of both theoretical properties and empirical performance?

---

> ### Author Response · Authors · 2024-11-23
> **Response to Reviewer vGg8 (Part 1)**
>
> Dear Reviewer,
>
> Thank you for your detailed and insightful comments. Below, we address your points and elaborate on the revisions made to the paper in response to your feedback.
>
> **1. Reliance on Gaussian mixture parameterization. <...> What makes Gaussian mixture parameterization particularly suitable for the proposed loss compared to alternative parameterizations? <...> The discussion of alternative parameterizations...**
>
> Dear reviewer, you accurately captured the essence of our parameterization approach in your summary, where you noted that it "allows *closed-form expressions* for their loss terms". Indeed, the Gaussian Mixture Model (GMM) parameterization offers a significant advantage by enabling the normalization constant $Z_{\theta}$ to be evaluated in closed form. This capability facilitates closed-form expressions for the loss terms, thereby improving both computational efficiency and clarity.
>
> We agree with you that the GMM parameterization may experience issues in scaling to high dimensions. To further elaborate your comment, we made two additional edits in the revised paper.
>
> - **First**, we have introduced new **Theorem 3.1** to demonstrate that our solver, using GMM-based parameterization, acts as a universal approximator. Under mild assumptions, it can approximate the true conditional distribution to an arbitrary degree of accuracy in terms of KL-divergence, provided that a sufficient number of Gaussian components are utilized. For further details, please refer to **Section 3.4** and **Appendix C.2** in the updated version of our paper.
>
> - **Second**, we demonstrated the possibility to enable the neural parameterization for both the potential function $f^\theta$ and the cost function $c^\theta$ within our framework. This neural parametrization may potentially allow our framework to handle tasks is higher dimensions as detailed in the newly added **Appendix A** in our revised paper.
>
> - **Third**, as we noted in the discussion section, we believe that our ideas have potential to be incorporated into advanced learning techniques, e.g., such as diffusion models. This is due to our framework being based on the well-established KL-divergence optimization principles. This serves as a promising avenue for future research.
>
> **2. How does the proposed method compare to existing semi-supervised domain translation techniques in terms of both theoretical properties and empirical performance?**
>
> **Theoretical comparison.** Our paper introduces a novel, unified framework for semi-supervised domain translation that seamlessly integrates both paired and unpaired data while providing strong theoretical guarantees. In contrast, existing methods that often rely on heuristically designed loss functions [1] or manually incorporate paired data into a conditional framework [2], our approach is grounded in solid theoretical principles.
>
> - Our loss function (Equation 12) is derived from a KL-divergence minimization framework. When this loss function converges, it ensures the recovery of the true conditional plan $\pi^\star$, as $\text{KL}(\pi^\star \Vert \pi^\theta) = 0$. To our knowledge, this is the first method to address the problem in this manner, offering a  principled theoretical foundation.
>
> - Our loss function is related to inverse entropic optimal transport (OT), as discussed in Section 3.2. This connection enables ML researchers to leverage a rich set of existing methods for solving semi-supervised domain translation tasks using the already well-established techniques from computational OT.
>
> **Empirical Properties \& Comparative Performance.** Following your comment about the experiments and the comments of other reviewers, we added various baselines to the real-world data experiment and revised **Section 5.2**. In particular, following the suggestion of the other reviewer Nafh, we added two additional semi-supervised baselines (CNF-SS and CGMM-SS, see **Appendix B.2.2**). *Our method consistently outperforms the alternatives in the considered experiment.*
>
> At the same, we understand your concern about comparison with the existing semi-supervised methods for image-to-image (I2I) translation [1,3]. Here we highlight two things.
> - **First**, our the goal of paper is to make the first step toward establishing theoretically-grounded methods for semi-supervised domain translation rather than design a large-scale approach for complex I2I tasks. As we already pointed above, there is a feasible potential to scaling our framework to large-scale problems via neural parameterization (**Appendix A**).
>
> - **Second**, our experimental evaluation already covers a kind of principal versions of losses which are used for I2I (e.g., Conditional and Unconditional GANs) and clearly demonstrates that they fail to learn the true conditional data distributions even in 2D.

---

> > ### Author Response · Authors · 2024-11-23
> > **Response to Reviewer vGg8 (Part 2)**
> >
> > **3. The authors describe their parameterization as "light-speed" - what are the specific computational advantages that justify this claim?**
> >
> > The term "light-speed" comes from the works Light-SB [4] and Light-SBM [5]. To enhance clarity, we have changed this term to "light" in the updated version of our paper. While our solver can be considered "light-speed," as it achieves full convergence on an Apple M1 CPU in approximately 8–10 minutes—consistent with the runtimes reported in these earlier studies—our main objective is to develop a fundamentally new framework for semi-supervised domain translation.
> >
> > **Concluding Remarks.** We kindly invite you to respond to our post and let us know if the clarifications provided adequately address your concerns regarding our work. If there are any remaining points, we would be happy to discuss them further during the review phase. Otherwise, if you find our responses satisfactory, we would greatly appreciate your consideration in raising your score.
> >
> > **References.**
> >
> > [1] Yang, Zaifeng, and Zhenghua Chen. "Learning from paired and unpaired data: Alternately trained CycleGAN for near infrared image colorization." In 2020 IEEE International Conference on Visual Communications and Image Processing (VCIP), pp. 467-470. IEEE, 2020.
> >
> > [2] Gu, Xiang, Liwei Yang, Jian Sun, and Zongben Xu. "Optimal transport-guided conditional score-based diffusion model." Advances in Neural Information Processing Systems 36 (2023): 36540-36552.
> >
> > [3] Xiaole Tang, Xin Hu, Xiang Gu, and Jian Sun. "Residual-conditioned optimal transport: Towards structure-preserving unpaired and paired image restoration". ICML 2024.
> >
> > [4] Korotin, Alexander, Nikita Gushchin, and Evgeny Burnaev. "Light Schr{\"o}dinger Bridge." In The Twelfth International Conference on Learning Representations.
> >
> > [5] Gushchin, Nikita, Sergei Kholkin, Evgeny Burnaev, and Alexander Korotin. "Light and Optimal Schr{\"o}dinger Bridge Matching." In Forty-first International Conference on Machine Learning. 2024.

---

> > > ### Author Response · Authors · 2024-11-28
> > > **Official Comment by Authors**
> > >
> > > Dear Reviewer vGg8,
> > >
> > > Thank you for the valuable feedback on our work. This is a gentle reminder that the discussion phase concludes in less than three working days, specifically at 11:59 PM AoE on December 2.
> > >
> > > We are eager to address any additional questions you may have. If you feel that our responses have adequately addressed your concerns, we kindly ask you to consider increasing your score.
> > >
> > > Your thoughtful comments and time are greatly appreciated.

---

> > > > ### Comment · Reviewer_vGg8 · 2024-12-01
> > > >
> > > > Thanks for the detailed clarification and added experiments. While these clarify some of my points, in my view, my main concern is still not adressed. Specifically:
> > > >
> > > > - "While the related work section discusses multiple approaches to semi-supervised domain translation, the paper lacks empirical comparisons against any of these existing methods. This makes it difficult to assess the relative advantages of the proposed approach compared to established methods that also leverage both paired and unpaired data."
> > > >
> > > > - "The paper would be substantially strengthened by: 1) clearly articulating the advantages of the proposed method over existing approaches, and 2) providing direct empirical comparisons against state-of-the-art semi-supervised methods. Currently, comparisons are limited to baseline methods that were not specifically designed for semi-supervised domain translation."
> > > >
> > > > The authors mention "many existing semi-supervised domain translation methods combine paired and unpaired data ... (Jin et al., 2019, M3.3), (Tripathy et al., 2019, M3.5), (Mustafa & Mantiuk, 2020, M3.2), (Paavilainen et al., 2021, M2), (Panda et al., 2023, Eq. 8), (Tang et al., 2024, Eq. 8)", but do not compare to any of them. While I appreciate the two added baselines, these are not SOTA semi-supervised domain translation methods and this comparison is still missing to me. This makes it difficult to assess the relative advantages of the proposed approach. Thus, I am unsure about the practical impact of the proposed method and choose to keep my score.

---

### Official Review · Reviewer_oBqy · 2024-11-10

**Soundness:** 2
**Presentation:** 2
**Contribution:** 2
**Rating:** 3
**Confidence:** 3

**Summary:**

This paper introduces a novel semi-supervised learning approach that leverages both paired and unpaired data to learn conditional distributions through a data likelihood maximization framework. The proposed method constructs a new loss function that integrates paired and unpaired data directly into the objective, with a connection to inverse entropic optimal transport (OT). By framing the problem as an inverse entropic OT, the authors develop an algorithm that employs Gaussian mixture parameterization to approximate the conditional distribution in a computationally efficient manner. The empirical evaluation demonstrates that this method can achieve effective learning outcomes even with limited paired data, as long as sufficient unpaired data is available, showing promising results on synthetic and real-world datasets.

**Strengths:**

Clarity: The paper is reasonably structured and provides some intuition on connecting OT with likelihood maximization.

Empirical Results: The empirical tests show potential for the method to handle limited paired data, a typical challenge in semi-supervised setups.

**Weaknesses:**

Justification of Loss Function: The paper lacks sufficient theoretical or empirical justification for the specific loss function design. The decomposition into paired and unpaired data components seems plausible, but more discussion on its necessity or optimality would strengthen the work. Additionally, while the paper claims the loss solves inverse entropic OT, the practical implications of this for general semi-supervised tasks are underexplored.

Limited Comparisons: The experimental section lacks comparisons with several prominent semi-supervised learning methods that do not rely on OT or Gaussian mixtures. Without comparisons to these methods, it is challenging to gauge whether the proposed approach offers an advantage over non-OT-based techniques.

**Questions:**

See weakness

---

> ### Author Response · Authors · 2024-11-23
> **Response to Reviewer oBqy**
>
> Thank you for taking the time to review our paper and provide us useful feedback. Below are the answers to your questions.
>
> **1. Justification of Loss Function: The paper lacks sufficient theoretical or empirical justification for the specific loss function design. The decomposition into paired and unpaired data components seems plausible, but more discussion on its necessity or optimality would strengthen the work.**
> Our loss is essentially the KL divergence with a smart energy-based reparameterization (equation 11) to be able to consider both paired and unpaired data.
> - The theoretical justification for **our loss function** is that it reaches the minimum if and only if the optimized distribution $\pi^{\theta}$ coincides with the true data distribution $\pi^{*}$ that we need to learn ($KL(\pi^{\star},\pi^{\theta})=0$ if and only if $\pi^{\star}=\pi^{\theta}$).
> - The theoretical justification for **our energy-based reparameterization** (11) through $E^{\theta}$ and its decoupling to $c^{\theta}$ and $f^{\theta}$ is that such parameterization is not any restrictive and can represent the true data distribution $\pi^{\star}(y|x)$. To achieve this optimality, term $c^{\theta}(x,y)$ in our parametrization can be set to $-\log \pi^{\star}(y|x)$, $f^{\theta}(y)=0$. In this case when $\varepsilon= 1$, from (9) one immediately gets that $Z_{\theta}(x)=1$. As a result, from (11) one derives $$\pi^{\theta}(y|x)=\exp(-E_{\theta}(y|x))/Z_{\theta}=\exp(-\log\pi^{\star}(y|x))=\pi^{\star}(y|x).$$ In practice, the considered parametric class for $c^{\theta},f^{\theta}$ may be insufficient to exactly represent $\pi^{\star}$. Nevertheless, in response to your question, we included a *new theoretical result* showing that our considered GMM-based parameterization can still minimize the loss up to any desired small error, see **Section 3.4** in the revision.
>
> **2. Additionally, while the paper claims the loss solves inverse entropic OT, the practical implications of this for general semi-supervised tasks are underexplored.**
> Thanks to our established connection, now ML researchers may be able to leverage a rich set of existing methods for computing OT problems for solving semi-supervised domain translation tasks.
> - **First**, our paper already demonstrates how recent ideas in computational OT based on Gaussian mixtures can be exploited to design a theoretically-justified algorithm for semi-supervised domain translation.
> - **Second**, answering the questions of the reviewers, we added a new **Appendix A** (in the revised version) which provides a discussion and a proof-of-concept experiment showing that other recent advances on solving OT with neural networks can also be used within our framework.
> - **Third**, we believe that our findings may motivate future researches to generalize our ideas proposed here to more general semi-supervised setups. This direction is a promising avenue for follow-up works.
>
> **3. The experimental section lacks comparisons with several prominent semi-supervised learning methods that do not rely on OT or Gaussian mixtures. Without comparisons to these methods, it is challenging to gauge whether the proposed approach offers an advantage over non-OT-based techniques.**
> Following your suggestion, we added various GAN and Normalizing Flow-based methods to the real-world data experiment and revised **Section 5.2**. In particular, following the suggestion of the other reviewer Nafh, we added two additional semi-supervised baselines (CNF-SS and CGMM-SS, see **Appendix B.2.2**). *Our method consistently outperforms the alternatives in the considered experiment.*
>
> **Concluding remarks.** Please respond to our post to let us know if the clarifications above suitably address your concerns about our work. We are happy to address any remaining points during the discussion phase; if the responses above are sufficient, we kindly ask that you consider raising your score.

---

> > ### Author Response · Authors · 2024-11-28
> > **Official Comment by Authors**
> >
> > Dear Reviewer oBqy,
> >
> > Thank you for the valuable feedback on our work. This is a gentle reminder that the discussion phase concludes in less than three working days, specifically at 11:59 PM AoE on December 2.
> >
> > We are eager to address any additional questions you may have. If you feel that our responses have adequately addressed your concerns, we kindly ask you to consider increasing your score.
> >
> > Your thoughtful comments and time are greatly appreciated.

---

> > > ### Author Response · Authors · 2024-12-03
> > > **Gentle Reminder**
> > >
> > > Dear Reviewer oBqy,
> > >
> > > Thank you again for taking the time to review our paper and provide your valuable feedback.
> > >
> > > As a reminder, the discussion period will conclude in less than 24 hours, specifically on December 2 (AoE). After this point, it will no longer be possible to submit additional comments by you.
> > >
> > > We would greatly appreciate any feedback you may have on our responses to your reviews. Should you have any remaining questions or concerns, we are more than happy to address them during the remaining discussion time.
> > >
> > > Thank you in advance for your input.

---

### Author Response · Authors · 2024-11-23
**General response and revision**

Dear Reviewers,

We sincerely appreciate your thoughtful comments and constructive feedback on our work. We are grateful for the time and effort each of you (oBqy, vGg8, Nafh) dedicated to reviewing our manuscript. We are particularly pleased that reviewers (vGg8, Nafh) acknowledged the significance of our novel maximum-likelihood framework, which combines paired and unpaired data within a theoretically grounded approach that requires minimal hyperparameters (as specifically noted by reviewer vGg8). Additionally, we are encouraged by the positive feedback from all reviewers (oBqy, vGg8, Nafh) regarding the connection between our proposed loss function and inverse entropic optimal transport, as well as the clarity of our paper’s exposition. For ease of review, all changes made in the revised manuscript are highlighted in **blue**.

We have thoroughly considered your suggestions and implemented substantial revisions to the manuscript, which address the following key points:

1. **Justification of Parametrization (oBqy, vGg8, Nafh):**

   To address concerns regarding our choice of Gaussian mixture parametrization, we have added **Theorem 3.1**, which demonstrates that this parametrization, under mild assumptions, can approximate a broad class of conditional plans with arbitrary accuracy. Detailed proofs supporting this result are now provided in **Appendix C.2**.

2. **Exploration of Advanced Parametrizations (oBqy, vGg8, Nafh):**

   To further emphasize the flexibility of our framework, we have included a new section in **Appendix A** that explores an implementation using an energy-based model. This model leverages a neural network to parametrize both the potential and cost functions, illustrating the applicability of our framework beyond Gaussian mixtures. We provided a proof-of-concept experiment in the Appendix A.

3. **Comparison with Alternative Loss Functions (Reviewer Nafh) and Baseline Models (oBqy, vGg8, Nafh):**

   To thoroughly evaluate our proposed loss function, we conducted additional experiments comparing it with an alternative loss suggested by Reviewer Nafh. These new results, presented in the revised manuscript, underscore the strengths of our approach. We have also expanded baseline comparisons from Section 5.1 (2D illustrative example) to include results in Section 5.2 (Weather experiments), providing a more comprehensive empirical evaluation.

We believe these revisions have substantially strengthened the manuscript and have addressed your concerns. **We have also updated our codebase to include all relevant baselines, ensuring reproducibility.**

Thank you once again for your valuable feedback and guidance. Detailed responses to your individual comments will be provided below in responses to each of your.

---

### Meta-Review · Area_Chair_E2uC · 2024-12-23

**Metareview:**

This paper proposes a semi-supervised approach for domain translation that combines paired and unpaired data within a likelihood maximization framework. The authors establish a connection between their proposed loss function and inverse entropic optimal transport, introducing a Gaussian mixture parameterization for computation. The method is evaluated on synthetic 2D data and a weather prediction task.

The paper's main strength lies in its theoretical approach to semi-supervised domain translation, integrating paired and unpaired data in a single objective. The connection to inverse entropic optimal transport is noteworthy.

However, the paper suffers from several significant weaknesses. The experimental validation is limited, especially for higher-dimensional problems, and lacks comparisons with state-of-the-art semi-supervised domain translation methods. The justification for the Gaussian mixture parameterization is insufficient, raising concerns about its scalability. The intuition and explanation for some aspects of the objective function are incomplete, and the practical motivation for scenarios with both paired and unpaired data is unclear.

The primary reasons for rejecting this paper are the limited experimental validation and the lack of comparisons with existing state-of-the-art methods. The practical impact and scalability of the proposed approach remain unclear, and the authors' responses did not sufficiently address these concerns. The reliance on Gaussian mixture parameterization raises questions about the method's applicability to more complex, high-dimensional problems. Furthermore, the insufficient explanation of the objective function's intuition makes it challenging to assess the method's full potential and limitations.

**Additional Comments On Reviewer Discussion:**

During the rebuttal period, reviewers raised concerns about the limited experimental validation, lack of comparisons with state-of-the-art methods, the intuition behind the objective function, the scalability of the Gaussian mixture parameterization, and the practical motivation for the proposed scenario.

The authors attempted to address these points by adding some comparisons, providing more explanations for their objective function, introducing a theoretical result on universal approximation, discussing potential neural parameterizations, and offering examples of relevant real-world scenarios. However, their responses were not fully satisfactory.

While some additional comparisons were added, they did not include state-of-the-art semi-supervised domain translation methods. The explanations for the objective function were expanded, but deeper intuition was still lacking. The new theoretical result on universal approximation was valuable, but practical concerns about scalability remained unresolved. The potential for neural parameterizations was promising but not fully explored in the current work.

Overall, while the authors made efforts to address the reviewers' concerns, the improvements were not sufficient to overcome the paper's weaknesses, particularly in terms of experimental validation and demonstration of practical impact. These unresolved issues support the decision to reject the paper.

---

### Decision · Program_Chairs · 2025-01-22

Reject